# Using metagenomics and whole-genome sequencing to characterize enteric pathogens across various sources in Africa

Cecilie Thystrup [1] ✉, Tesfaye Gobena[2], Elsa Maria Salvador[3], Olanrewaju Emmanuel Fayemi [4], Happiness Kumburu[5,6], Elna M. Buys[7], Josphat Gichure[7], Belisário T. Moiane[3], Dinaol Belina [8], Ephrasia A. Hugho [5,9], Sara Faife[10], Tosin Segun Ogunbiyi[4], Gabriel Akanni [4], Christianah I. Ayolabi[4,11], Blandina Mmbaga [5,9], Kate M. Thomas [12], Sara M. Pires[1], Patrick Murigu Kamau Njage[1] & Tine Hald[1]

Foodborne diseases (FBDs) remain a major public health concern in low- and middle-income countries (LMICs), with the African region carrying the heaviest burden globally. Surveillance efforts in these settings often overlook rural and resource-limited communities, limiting our understanding of pathogens transmission dynamics in these settings. In this study, we use whole-genome sequencing (WGS) and metagenomic approaches to characterize enteric pathogens from human, animal, and environmental sources across four African LMICs between 2019 and 2023. We analyze 446 bacterial isolates of *Salmonella*, *Shigella*, *Escherichia coli*, and *Campylobacter*, of which 380 high-quality genomes were subjected to phylogenetic and genotypic analyses. Additionally, 139 of 168 metagenomic samples pass quality control and were assessed for pathogen abundance and diversity. Our results reveal a geographically stable distribution of foodborne pathogens over time, suggesting persistent ecological or infrastructural factors influencing their maintenance. Genomic comparisons also identify closely related isolates across distinct sources and regions, pointing to potential transmission routes. These findings highlight the value of incorporating targeted environmental and food-chain sampling into surveillance strategies and demonstrate that metagenomic sequencing can serve as a practical and informative addition to WGS-based surveillance in resource-limited settings.

Foodborne diseases (FBD) represent a critical public health issue that often results in severe health complications and substantial economic burden[1,2]. The impact of FBD is especially large in low- and middle-income countries (LMICs), where the African region bears the highest burden with more than a billion cases annually[3,4]. FBDs such as norovirus infection and salmonellosis are mostly managed efficiently in high-income countries but continue to persist in LMICs due to

socioeconomic and environmental drivers (including limited financial resources, insufficient and/or inadequate infrastructure, lack of trained personnel, and competing public health priorities)[5].

The primary means of acquiring an infection with a pathogen commonly transmitted via food is through the consumption of contaminated food or water, as well as direct contact with infected individuals or animals[6]. Among the common foodborne pathogens (FBPs)

in Africa, enteric pathogens such as non-typhoidal *Salmonella* spp. (NTS), *Campylobacter* spp., pathogenic *Escherichia coli*, and *Shigella* spp. are particularly prevalent[6,7].

To address the challenges of FBDs, several intervention strategies like improved food safety practices, robust public health policies, and effective disease surveillance systems need to be implemented[8,9]. However, the current understanding of FBD epidemiology and dynamics in LMIC needed to inform these strategies is limited[5]. Efforts to advance disease surveillance in LMICs, including identifying and managing FBD-causing pathogens, often fail to reach communities in rural areas, where a lack of infrastructure and resources limits the effectiveness of a surveillance system[10]. The absence of robust surveillance and monitoring systems in LMICs also impedes efforts to detect, prevent, and contain FBD outbreaks, thereby aggravating the public health burden and undermining progress towards achieving global health security[11–13]. Therefore, there is a critical need for comprehensive data to better understand the scale of the diseases and their transmission between different reservoirs[14]. To determine the most effective interventions to mitigate the risks, key questions about the specific pathogens involved, their transmission routes, and reservoirs need to be answered[3].

Increased use of whole-genome sequencing (WGS) in outbreak detection impacts food safety by allowing for a more precise approach to pathogen detection, characterization, and identification[15,16]. For instance, genomic monitoring and the application of WGS have been used for outbreak detection and investigation of foodborne diseases in high-income countries[17–19]. One example is the investigation of an outbreak of *Salmonella Typhimurium* ST34, which was linked to chocolate products and spanned multiple countries in the EU/EEA in 2022[20]. Other studies have also shown the usefulness of applying metagenomic data to find correlations between health- and environmental factors across multiple countries[21]. The practicality of applying these approaches for surveillance of foodborne infections to detect outbreaks has rapidly increased with the reduced cost of WGS and metagenome sequencing, which allows for the detection of new patterns across sources[13].

By using a combination of genomic data, like WGS or metagenomic sequences, with epidemiological data, it is possible to improve the detection and characterization of foodborne pathogens and thus help facilitate the establishment of an effective FBD surveillance system in Africa. The use of both WGS and metagenomic analysis methods could provide a better understanding of the distribution, diversity, and dynamics of the target pathogens across multiple sources in LMICs, each method providing different levels of detail. This knowledge could then inform public health interventions and strategies for reducing the burden of FBDs in these regions.

The "Foodborne Disease Epidemiology, Surveillance and Control in African Low- and Middle-Income Countries" (FOCAL)-project aimed to explore the potential of using genomic data together with epidemiological data to build an FBD surveillance system in African LMIC. The project linked public-health surveillance data with genomic data from food, animals, and the environment to provide the best evidence of the health impact of, and the relative contribution of different sources, for FBD in African LMIC.

The aim of this study was to provide an overview of the diversity, abundance, and genomic relationship of foodborne pathogens in four African LMIC using WGS and metagenomic data applied in human, animal, food, and environmental samples, collected as part of the FOCAL project. Furthermore, the study also aimed to assess the potential of metagenomic sequencing as a complementary tool for foodborne disease surveillance in resource-limited settings.

## Results

### Pathogen isolate overview and data quality

A total of 3417 samples were collected between December 2019 and March 2023 from 24 different sources in Ethiopia, Nigeria, Mozambique, and Tanzania. Of these, a subset underwent culture and subsequent isolation and WGS, whereas a different subset was submitted for metagenomic sequencing. A total of 446 isolates of the target pathogens were recovered from the samples (Table 1), representing a range of reservoirs, including human ($n = 244$) and animal ($n = 37$) reservoirs, food products ($n = 89$), water ($n = 45$), and waste ($n = 31$).

Following quality assessment of assembled genomes, 380 of the initial 446 pathogen isolates passed thresholds for inclusion in downstream analysis. The final dataset included *E. coli* ($n = 207$), *Shigella* spp. ($n = 11$), *Salmonella* spp. ($n = 138$), and *Campylobacter* spp. ($n = 24$). Among these high-quality genomes, the number of contigs ranged from 20 to 4840 (median: 111), with N50 values spanning 2759 to 728,755 bp (median: 177,204 bp).

Isolates were characterized by MLST and serotype prediction (Supplementary Data 1). MLST profiles were obtained for 77% (160/207) of *E. coli*, 90% (124/138) of *Salmonella* spp., and 92% (22/24) of *Campylobacter* spp. isolates. *E. coli* isolates were assigned to 83 different STs, with ST13857 ($n = 17$), ST38 ($n = 8$), and ST131 ($n = 7$) among the most frequent. *Salmonella* spp. isolates were dominated by ST1208 ($n = 23$), followed by ST93 ($n = 7$), ST49 ($n = 7$) and ST808 ($n = 6$). *Campylobacter* showed considerable sequence type diversity, with ST19 ($n = 5$) as the most common.

Serotype predictions revealed limited overlap in antigenic profiles across geographic regions. Among *E. coli* isolates, the most frequently observed serotypes were O4:H12 ($n = 8$) and O113:H4 ($n = 8$), all originating from Ethiopia ($n = 14$) and Mozambique ($n = 4$). Approximately 10% (21/207) of *E. coli* isolates could not be assigned a serotype. For *Salmonella*, the most prevalent serovar, 42:r:- ($n = 20$), was detected exclusively in samples from Tanzania. Other prevalent serotypes found were *S*. Natal ($n = 9$ from Nigeria, *S*. Eastbourne ($n = 6$), *S*. Gaminara ($n = 6$), *S*. Kottbus ($n = 6$), and *S*. Muenchen ($n = 6$), all from Ethiopia.

Annotation with the ABRIcate pipeline revealed a heterogeneous distribution of pathogenic traits across the isolate collection. The most prevalent virulence gene was *astA*, identified in 51 isolates (24.6%, 51/207). This gene encodes the enteroaggregative heat-stable toxin 1 (EAST1) and is broadly distributed among multiple diarrheagenic *E. coli* pathotypes like enteroaggregative (EAEC), enterotoxigenic (ETEC), Shiga toxin-producing (STEC), and enterohemorrhagic (EHEC) strains[22]. In addition, a small number of isolates contained *bfp* (bundle-forming pilus) and *eae* (intimin), which are typically associated with enteropathogenic *E. coli* (EPEC) and STEC, respectively[23,24].

### Taxonomic composition of metagenomic samples

Between December 2019 and March 2023, 168 metagenomic samples from 13 different sources were collected across the four African countries enrolled in the project. Sources included children with diarrhea, their caretakers and siblings, animals in their household vicinity, including poultry and ruminants, as well as samples of wastewater samples and sewage. Sequencing yielded between 43.8 and 145.2 million paired-end reads per sample (median: 60.2 million) before pre-processing. Following quality control and exclusion of samples with low DNA content, 139 samples were retained for downstream analyses (Table 2). Sequencing yielded between 42.1 and 138.3 million paired-end reads per sample (median: 58.0 million) after pre-processing, corresponding to a sequence output ranging from 6.11 to 20.35 Gb per sample, with a median of 8.45 Gb. This provided sufficient sequencing depth for downstream taxonomic profiling, diversity analyses, and metagenome assembly.

Across samples, between 1105 and 4500 (median: 3014) genera were detected. Top 15 genera accounted for 63.57% of the total abundance across all samples (Fig. 1). Abundant taxa included *Escherichia* (13.21%), *Enterococcus* (9.01%), *Bifidobacterium* (6.46%), and *Lactobacillus* (4.15%), reflecting both human- and environment-associated microbial communities. Microbial community

**Table 1 | Overview of the whole-genome sequencing (WGS) isolates collected from the four countries (Ethiopia, Mozambique, Nigeria, and Tanzania) in the project (n = 446)**

| | | Ethiopia | | | | Mozambique | | | Nigeria | | | Tanzania | | |
|---|---|---|---|---|---|---|---|---|---|---|---|---|---|---|
| | | Campylobacter spp. | Escherichia coli | Salmonella spp. | Shigella spp. | Escherichia coli | Salmonella spp. | Shigella spp. | Escherichia coli | Salmonella spp. | Shigella spp. | Escherichia coli | Salmonella spp. | Shigella spp. |
| Human | Case | 4.3% (13/299) | 15.4% (46/299) | 6.7% (20/299) | 1% (3/299) | 56.9% (58/102) | – | (2/102) | 8.7% (26/300) | 6.7% (20/300) | 0.33% (1/300) | 6.4% (18/281) | 2.5% (7/281) | (1/281) |
| | Caretaker | 4.6% (4/87) | 18.4% (16/87) | 5.7% (5/87) | 2.3% (2/87) | – | – | – | – | – | – | 1% (1/99) | 1% (1/99) | – |
| Animals | Bovine/cattle | 1.1% (2/188) | 1.1% (2/188) | 3.2% (7/188) | – | – | – | – | – | – | – | – | – | – |
| | Chicken/hens | 2.1% (4/188) | 4.3% (8/188) | 3.2% (6/188) | – | – | – | – | – | – | – | – | 2.3% (4/174) | – |
| | Dog | – | – | – | – | – | – | – | – | – | – | – | 0.6% (1/174) | – |
| | Goat | 1.1% (2/188) | – | – | – | – | – | – | – | – | – | – | 0.6% (1/174) | – |
| Food | Bovine meat | 1.5% (3/200) | 3% (6/200) | 5.5% (11/200) | 0.5% (1/200) | 3.4% (6/175) | – | – | – | 6.3% (7/112) | – | 0.5% (2/373) | 2.4% (9/373) | – |
| | Sheep/mutton meat | – | – | 0.5% (1/200) | – | – | – | – | – | – | – | – | – | – |
| | Goat meat | – | – | – | – | – | – | – | 3.6% (4/112) | 2.7% (3/112) | – | – | – | – |
| | Milk (cow) | 0.5% (1/200) | 0.5% (1/200) | – | – | 0.6% (1/175) | – | – | – | – | – | – | 0.8% (3/373) | – |
| | Pork | – | – | – | – | – | – | – | – | – | – | – | 0.3% (1/373) | – |
| | Other food | 1% (2/200) | 2.5% (5/200) | 1.5% (3/200) | – | 3.4% (6/175) | 0.6% (1/175) | 0.6% (1/175) | 0.9% (1/112) | 0.9% (1/112) | 0.9% (1/112) | – | – | – |
| | Fermented dairy (cow) | – | – | – | – | – | 0.6% (1/175) | – | – | – | – | – | – | – |
| | Fermented dairy (goat) | – | – | – | – | 1.7% (3/175) | 0.6% (1/175) | – | – | – | – | – | – | – |
| | Porridge | – | – | – | – | 1.1% (2/175) | 0.6% (1/175) | – | – | – | – | – | – | – |
| Water | Drinking water | – | – | – | 2.4% (1/41) | 10.8% (11/102) | 0.6% (1/175) | – | – | – | – | – | 10.5% (23/220) | – |
| | Other water | – | – | 2.4% (1/41) | – | – | – | – | – | – | – | – | 3.6% (8/220) | – |
| Waste | Other wastewater | – | 1.4% (1/70) | – | – | – | – | – | – | – | – | – | – | – |
| | Waste (animals) | 2.9% (2/70) | 10% (7/70) | 21.4% (15/70) | – | – | – | – | – | 1.1% (1/92) | – | – | – | – |
| | Waste (humans) | 5.7% (4/70) | – | 1.4% (1/70) | – | – | – | – | – | – | – | – | – | – |
| Total | | 4.2% (37/885) | 10.6% (92/885) | 7.7% (70/885) | 0.8% (7/885) | 15.7% (87/553) | 0.9% (5/553) | 0.5% (3/553) | 3.7% 31/827 | 3.9% (32/827) | 0.2% (2/827) | 1.8% (21/1152) | 5% (58/1152) | 0.1% (1/1152) |

composition varied by sample type and geographic origin. Among diarrheal cases, those from Ethiopia were enriched in *Bacteroides*, *Faecalibacterium*, and *Bifidobacterium*, while samples from Mozambique and Nigeria showed higher relative abundances of *Klebsiella* and *Escherichia*. In contrast, case samples from Tanzania were dominated by *Streptococcus* and *Bifidobacterium*. Notably, *Streptococcus* was also abundant in the two samples collected from asymptomatic children in Tanzania. Animal samples also showed distinct profiles. Chicken and hen samples from both Ethiopia and Tanzania exhibited a high relative abundance of *Lactobacillus*, while samples from Tanzania in particular contained higher levels of *Enterococcus*. Sewage samples displayed the highest overall taxonomic diversity, with *Pseudomonas* and *Enterobacter* consistently detected across all countries.

### Microbiome composition and source signatures
PCA of CLR-transformed species-level abundance data revealed distinct clustering patterns by sample source (Fig. 2). PC1 and PC2

accounted for 11.9% and 10.5% of the total variance, respectively. Diarrheal case samples tended to cluster along the PC2 axis (approximately −40 to 60), while samples from bovine/cattle and caregivers also showed a compact distribution, primarily within the 0–50 range on PC2. Sewage samples formed a distinct cluster, separated from other sources, occupying the 0–60 range on both PC1 and PC2 axes. Arrows representing the top 10 contributing taxa were projected into the ordination space, with most pointing in the direction of PC2, indicating these species were primary drivers of variation. Notable contributors included *Prevotella hominis*, *Prevotella copri*, *Bacteroides wexlerae*, *Anaerobutyricum hallii*, *Eubacterium rectale*, and *Ruminococcus bromii*, many of which are associated with the human or animal gastrointestinal microbiome[25–28]. PERMANOVA analysis confirmed a significant effect of source on community structure ($p = 0.0001$, $R^2 = 0.333$), consistent with the clustering observed in the ordination space.

### Temporal trends in FBD-causing pathogens in sewage samples
The overall pathogen load normalized by the number of samples per group showed a marked increase in 2021 across all countries (Fig. 3). Despite this rise, the relative abundance pattern among the four target pathogens remained largely consistent across years and countries. *E. coli* was consistently the pathogen with the highest abundance, while *Campylobacter* spp., *Salmonella* spp., and *Shigella* spp. remained less abundant. The estimated relative abundance of the four pathogens of interest also showed that the relative composition of these pathogens remained relatively stable. These findings suggested a temporal increase in total FBD pathogen burden in urban and rural sewage environments, particularly in 2021. However, the relative contribution of each pathogen to the total community remained broadly consistent, indicating that no single species disproportionally drove the observed increase in the sewage samples.

### Hierarchical clustering
A clustered heatmap of CLR-transformed abundances for the 50 most variable species revealed three major clusters among the source-country combinations (Fig. 4). While clustering did not strictly follow host reservoir, partial structure was observed, with human-associated samples (cases and caretakers) clustering together, and avian samples (chickens and ducks) forming internally coherent groups. Interestingly, some clustering appeared to reflect geographic rather than ecological similarity, for example, bovine

**Table 2 | Overview of the metagenomic samples collected from the four countries (Ethiopia, Mozambique, Nigeria, and Tanzania) in the project (*n* = 139)**

| | | Ethiopia | Nigeria | Mozambique | Tanzania | Total |
|---|---|---|---|---|---|---|
| Human | Case | 9 | 15 | 19 | 16 | 59 |
| | Caretaker | 3 | – | – | 8 | 11 |
| | Sibling | 1 | – | – | 2 | 3 |
| Animal | Bovine/cattle | 2 | – | – | 6 | 8 |
| | Pig | – | – | – | 1 | 1 |
| | Chicken/hens | 2 | – | 3 | 5 | 10 |
| | Duck | – | – | 4 | 2 | 6 |
| | Goat | 3 | – | – | – | 3 |
| | Sheep | 2 | – | – | – | 2 |
| | Donkey | – | – | 1 | – | 1 |
| Waste | Waste (human) | 3 | – | – | – | 3 |
| | Waste (animal) | 2 | – | – | – | 2 |
| | Sewage | 9 | 5 | 9 | 7 | 30 |
| Total | | 36 | 36 | 20 | 47 | 139 |

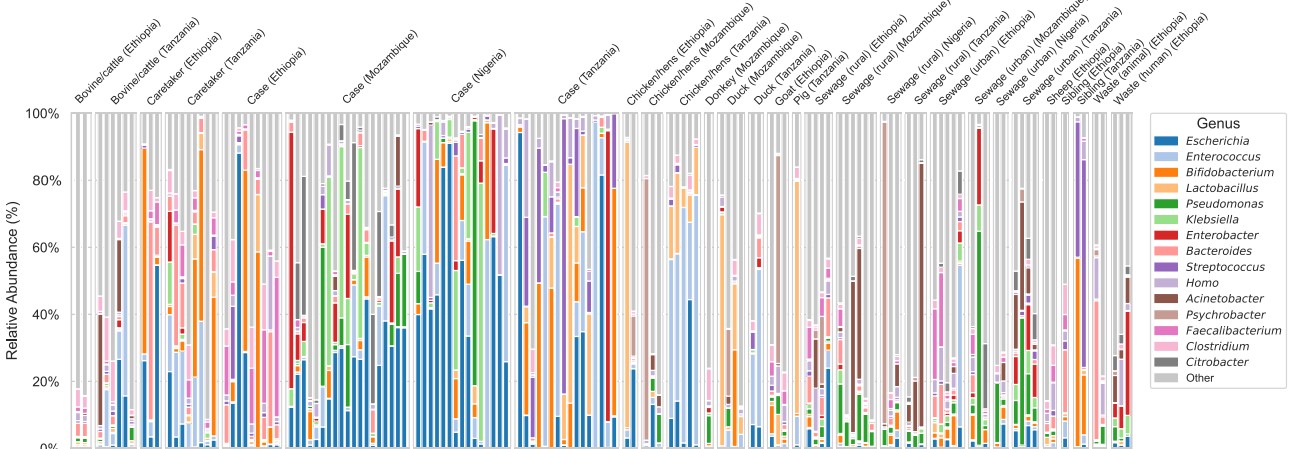

**Fig. 1 | Genus-level composition of microbial communities across sample sources and countries.** Stacked bar plots showing the relative abundance of the 15 most abundant bacterial genera across all metagenomic samples (*n* = 139), stratified by sample source (e.g., human, animal, environmental) and country.

Taxonomic classification was performed using Kraken2 and Bracken, and genus-level abundance was estimated from read counts. Genera not among the top 15 in relative abundance were grouped into an "Other" category for clarity. Source data are provided as a Source Data file.

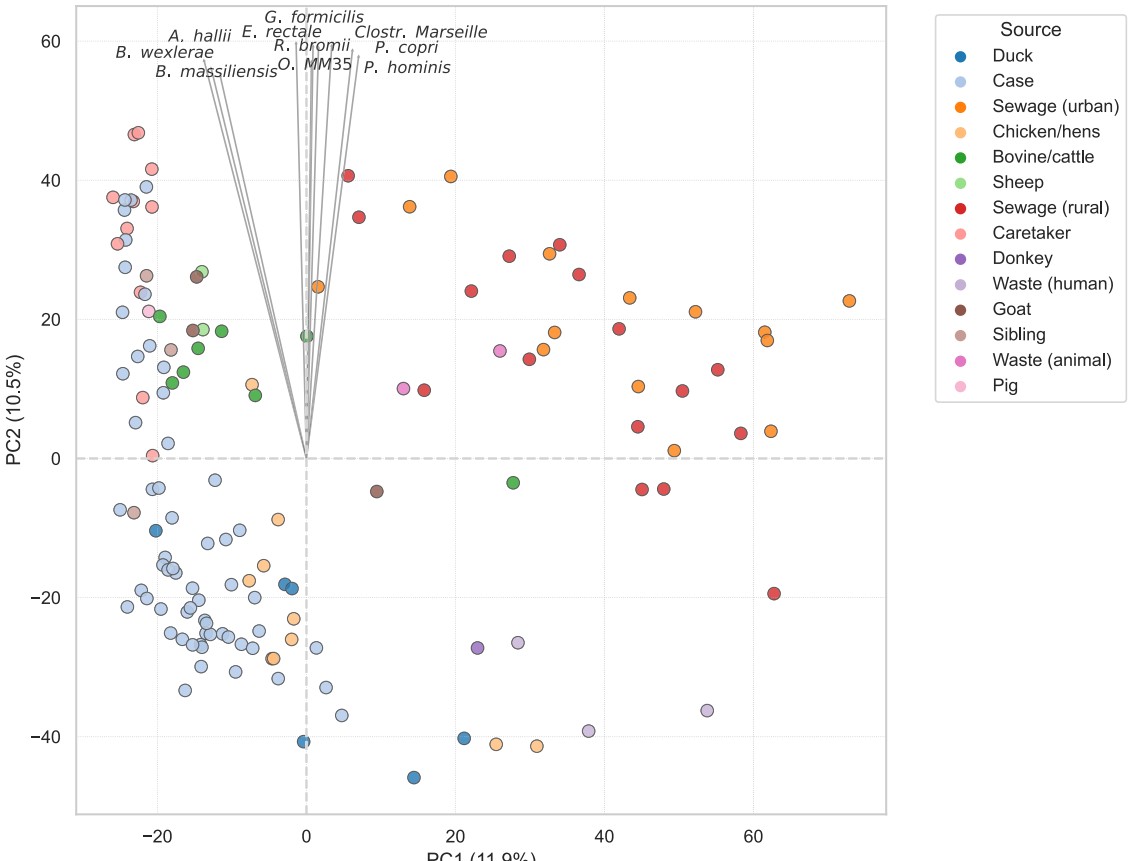

**Fig. 2 | Principal component analysis (PCA) of species-level microbial composition across 139 metagenomic samples.** Biplot showing the ordination of metagenomic samples based on centered log ratio-transformed (CLR) species-level abundance data. PCA was performed to explore variation in microbial community structure across sample sources and countries. Each point represents an individual sample, colored by source type. Arrows represent the top 10 species contributing most to the ordination space, with arrow direction indicating the gradient of increasing abundance and arrow length corresponding to the strength of contribution to the principal components.

samples from Tanzania clustered more closely with Tanzanian sewage samples than with bovine samples from Ethiopia. Notably, human-associated samples exhibited elevated abundance of *Bifidobacterium* spp., including *B. catenulatum* and *B. breve*, species commonly associated with the human gastrointestinal microbiome[29]. These taxa were consistently enriched in both cases and caretakers, but were largely absent from animal reservoirs such as poultry and cattle. This pattern suggests limited microbial overlap between human and non-human reservoirs, at least with respect to these key taxa.

### Recovery and characterization of high-quality MAGs

Assessment of assembled metagenomes using QUAST identified *E. coli*-identified contigs in 43 samples, *Salmonella* spp. in 2 samples, and *Campylobacter* spp. in a single sample. A total of 13 high-quality metagenome-assembled genomes (MAGs) were recovered, consisting of 12 *E. coli* MAGs and one Campylobacter MAG, all meeting MIMAG criteria for high-quality genomes (>90% completeness, <5% contamination). A summary of the 13 high-quality MAGs is provided in Table 3. *E. coli* MAGs were primarily recovered from children with diarrhea, with two exceptions, representing all four countries of the study. Genomic completeness ranged from 89.38% to 100%, with contamination levels below 2.35%. Pairwise comparison using FastANI to identify close correspondence between MAGs and cultured *E. coli* isolates with ANI values exceeded 96.5% in all cases, but were below 99.9%, indicating that strains detected by culture and isolation could not be linked to the MAGs. MAG genome sizes ranged from 3.7 to

4.51 Gb. MLST typing was successful for 6 *E. coli* MAGs, revealing STs overlapping with those observed in cultured isolates (Supplementary Data 1). Functional annotation identified between 18 and 79 virulence-associated genes per *E. coli* MAG; however, none were characteristic of canonical diarrheagenic *E. coli* pathotypes (*eae, stx1/2, ehxA, astA, aggR, bfp, ipaH*)[30].

### Phylogenetic analysis

The SNP-based phylogenetic tree constructed from high-quality *E. coli* genomes, including both MAGs and WGS isolates, revealed clear evidence of genomic clustering across sources and strain backgrounds. MAGs clustered alongside isolates, indicating that the genomes recovered from metagenomic data reflected at least some of the same population structure as observed among the cultured isolates (Fig. 5). One MAG, lacking an assigned sequence type, clustered within a clade composed entirely of ST131 isolates, a strain which is clinically relevant as it is the predominant *E. coli* lineage among extraintestinal pathogenic *E. coli* (ExPEC), suggesting phylogenetic relatedness despite the absence of a ST profile[31]. More broadly, isolates from diarrheal cases are frequently grouped into clades composed predominantly of other genomes from similar cases, including MAGs, highlighting potential epidemiological linkage or shared reservoirs.

In other instances, isolates and MAGs from similar sources, such as bovine meat and cattle, also clustered together, indicating source-associated genomic similarity. Despite this, the distribution of ST profiles was highly diverse and did not show a consistent pattern of overlap across countries or sources, suggesting that while local

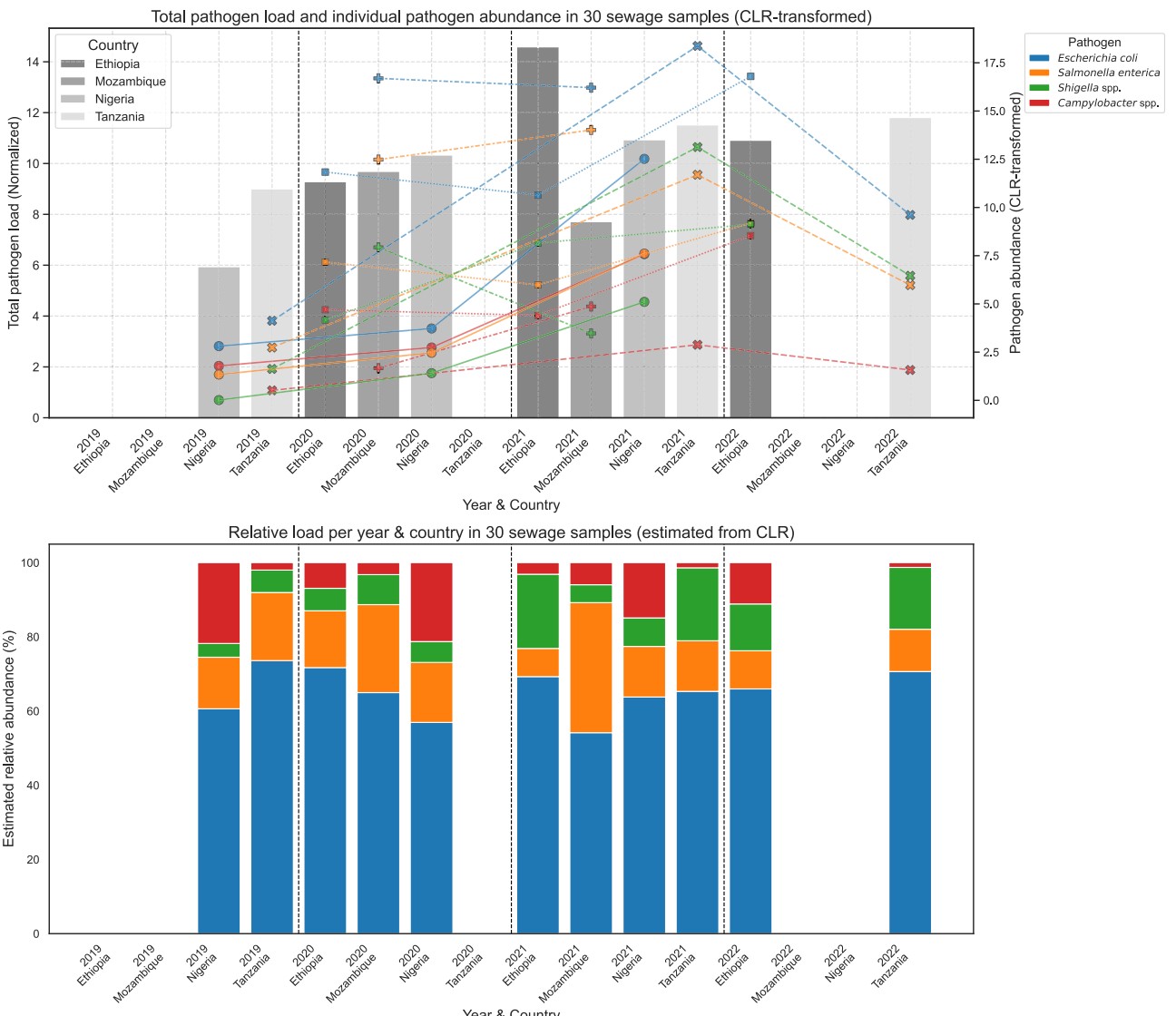

**Fig. 3 | Temporal trends in abundance and relative composition of foodborne pathogens in sewage samples from four African countries.** (Top) Bar plot showing the total normalized abundance of four key foodborne pathogens (*Escherichia coli*, *Salmonella enterica*, *Shigella* spp., and *Campylobacter* spp.) in sewage samples collected between 2019 and 2022, stratified by country and year. Total pathogen load was calculated by summing centered log ratio-transformed (CLR) abundance values across all four pathogens and normalizing by the number of samples per group. Overlaid line and scatter plots represent CLR-transformed abundance for each individual pathogen, illustrating changes in abundance across time and geography. (Bottom) Stacked bar plot of estimated relative abundances (%) of the same pathogens, calculated by applying an exponential transformation to CLR values and normalizing within each group. Source data are provided as a Source Data file.

clustering did exist in these samples, the broader *E. coli* population is highly genetically heterogeneous.

## Discussion

This study provides a genomic characterization of FBPs across four African LMICs using WGS and metagenomic sequencing data. Through the use of genomic isolate data from human, animal, food- and environmental samples, we provide valuable insights into the diversity, abundance, and subgroup relationships of key FBD-causing pathogens —*E. coli*, *Salmonella* spp., *Shigella* spp., and *Campylobacter* spp.—across different ecological and geographical settings. Analyses of metagenomic samples from the same pool, while not paired with the WGS dataset, provided a broader perspective on microbial community composition across regions and through time. This study fills a critical gap in the surveillance area of sub-Saharan Africa, where limited genomic data have historically limited pathogen tracking and surveillance.

The WGS analysis revealed that children with diarrhea, bovine/cattle, and water sources were the most common reservoirs of *E. coli* and *Salmonella* spp. isolates, highlighting these as potential sources of FBD transmission in the region. The sequence typing and serotyping results showed considerable heterogeneity across countries, with little overlap in STs or serovars for any of the targeted pathogens. This high diversity suggests that local ecological and epidemiological patterns shape FBD transmission in each country, making it difficult to generalize pathogen profiles or predict risk based solely on geography or sample type. Among *E. coli* isolates, several clinically relevant STs were detected. Notably, ST131 and ST38 were found across multiple countries and sources. ST131 is a well-known lineage among urinary tract infections and extraintestinal pathogenic *E. coli* (ExPEC), while ST38 has been less frequently reported in surveillance studies[31–34]. Four out of eight ST38 isolates also carried the virulence gene *astA*, consistent with findings from other studies reporting the presence of uropathogenic *E. coli* (UPEC)-like and EPEC-associated virulence genes in this

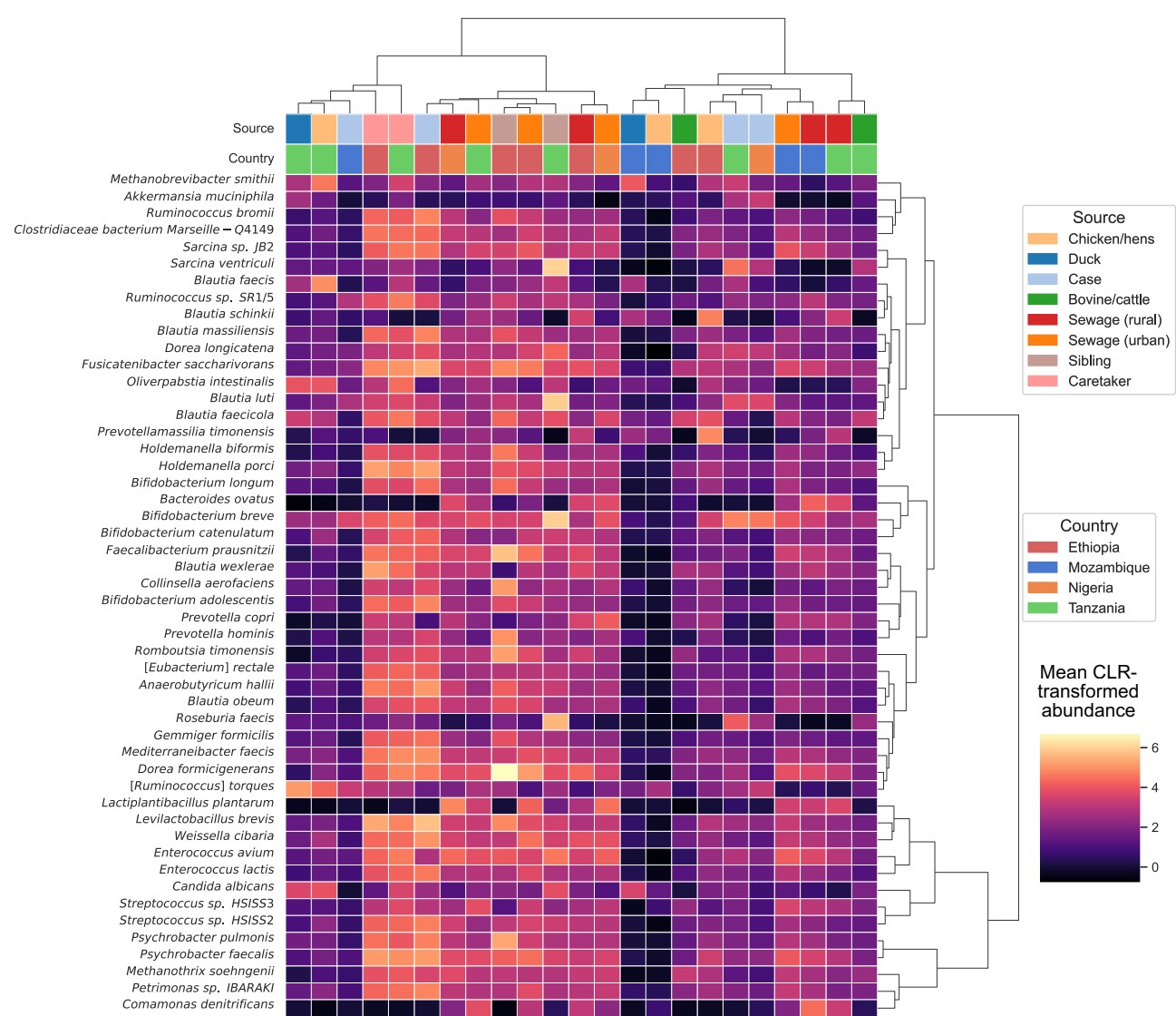

**Fig. 4 | Clustered heatmap of centered log ratio-transformed (CLR) abundances for the 50 most variable microbial species across source-country combinations.** Species with the highest variability across all samples were selected based on the standard deviation of CLR-transformed abundance values. Abundance values were aggregated by sample source and country using the mean CLR-transformed value. Source-country groups represented in only one country were excluded. Hierarchical clustering was performed independently on both rows (source-country combinations) and columns (species), using Ward's linkage and Euclidean distance.

lineage[32,35,36]. These results suggest that ST38 may warrant further investigation in the African context.

In contrast, ST13857 and ST1141, both found in Ethiopian samples, have not previously been associated with any *E. coli* infection in humans. Our results showed no diarrhea-associated virulence genes in ST1141 isolates and only limited virulence gene carriage in ST13857, suggesting these lineages may be non-pathogenic or opportunistic colonizers in their current ecological context. Nevertheless, their detection here highlights the limitations of global reference datasets, which are often biased toward high-income settings, and highlights the need to expand genomic surveillance efforts in underrepresented regions.

Among *Salmonella* spp. isolates, the most prevalent ST was ST1208, identified primarily in Tanzanian samples from children and drinking water, with a smaller number found in Nigerian meat samples. This ST has been previously reported in vegetable supply chains in South Africa and in both human and meat samples in Kenya, supporting its circulation across multiple African countries[37,38]. Other STs observed—such as ST93, ST49, and ST82—have not been reported in

any studies reporting *Salmonella* infections. The absence of studies reporting ST82 could be attributed to its primary identification in bovine and cattle samples from Ethiopia, which is possibly underrepresented in existing studies. Since ST93 and ST49 were detected in children with diarrhea, this suggests that their association with diarrhea is at least partly supported by the results of this study.

Importantly, we observed limited overlap between STs and predicted serovars in the isolates and the MAGs, particularly for *E. coli* and *Salmonella*. This may reflect the limitations of in silico serotyping, especially when applied to genomes from genetically diverse or undersampled regions. Closely related serovars may differ by only minor O- or H-antigenic features that are challenging to resolve using current databases and algorithms[39]. Misclassification is also possible when antigenic determinants are absent or highly divergent from those represented in curated reference datasets. The inability to assign STs or serovars to a subset of isolates further emphasizes the lack of representative African genomes in current typing databases and highlights the need to expand the genomic reference data to better reflect global diversity.

**Table 3 | Summary statistics for high-quality metagenome-assembled genomes (MAGs) identified in this study**

| MAG ID | Sample ID | Country | Source | Taxonomy (GTDB-Tk) | Completeness, Contamination (%) | Closest isolate (ANI%) | Genome size (Gb) | Virulence gene count (VFDB) | N50 | MLST (Achtmann scheme) |
|---|---|---|---|---|---|---|---|---|---|---|
| bin.18 | 43501121 | Tanzania | Case | E. coli | 95.36, 0.94 | 32881121 (96.65) | 4.39 | 70 | 23,453 | 648 |
| Bin.20 | 42781121 | Tanzania | Case | E. coli | 100, 2.35 | 31531121 (95.75) | 4.34 | 52 | 17,732 | Unknown[a] |
| Bin.26 | 3411121 | Mozambique | Case | E. coli | 93.31, 0.38 | 15481354-A (96.76) | 4.2 | 48 | 24,760 | Unknown[a] |
| Bin.2 | 4286121 | Tanzania | Case | E. coli | 90.76, 0 | 33321121 (96.83) | 3.7 | 18 | 28,001 | 152 |
| Bin.10 | 33791121 | Mozambique | Case | E. coli | 99.99, 0.02 | 15481354-A (96.68) | 4.51 | 39 | 153,185 | 101 |
| Bin.28 | 2105121 | Nigeria | Case | E. coli | 92.46, 1.46 | 31531121 (96.55) | 4.32 | 72 | 27,956 | Unknown[a] |
| Bin.64 | 13821121 | Ethiopia | Case | E. coli | 99.99, 0.21 | 33321121 (96.66) | 4.47 | 50 | 36,783 | 5614,3375[b] |
| Bin.8 | 21101121 | Nigeria | Case | E. coli | 90.09, 0.47 | 31531121 (96.5) | 4.07 | 60 | 21,843 | Unknown[a] |
| Bin.53 | 12851354 | Ethiopia | Chicken | E. coli | 93.3, 0.06 | 43951121 (96.9) | 4.07 | 39 | 37,009 | 10 |
| Bin.13 | 42761121 | Tanzania | Case | E. coli | 97.88, 1.09 | 15481354-A (96.80) | 4.16 | 39 | 68,804 | Unknown[a] |
| Bin.16 | 3145121 | Mozambique | Case | E. coli | 94.4, 0.67 | 21521121 (96.8) | 4.15 | 32 | 27,540 | Unknown[a] |
| Bin.22 | 42101123 | Tanzania | Sibling | E. coli | 93.74, 1.64 | 31121239 (96.7) | 4.36 | 32 | 32,976 | Unknown[a] |
| Bin.9 | 42951121 | Tanzania | Case | C. jejuni | 89.38, 1.42 | – | 1.55 | 79 | 9797 | 1458 |

Columns include country and sample source, taxonomic assignment using the Genome Database Taxonomy Toolkit (GTDB-Tk), genome completeness and contamination, closest matching isolate based on average nucleotide identity (ANI, determined via FastANI), genome size (in base pairs), number of detected virulence genes identified using ABRicate, found in the Virulence Factor Database (VFDB), N50 value (assembly continuity), and multi-locus sequence type (MLST; Achtman scheme).
[a]Partial match on several ST profiles.
[b]Nearest profile.

Metagenomic profiling of the samples provided additional resolution into the structure and distribution of microbial communities in samples from the four study countries. As expected, given the fecal origin of many samples, *Escherichia* and *Pseudomonas* were consistently abundant in the samples. This result is consistent with previous metagenomic studies of gut- and sewage-associated microbiomes and further validates the representativeness of the samples collected in this study[40,41]. Country-level differences were also observed in the taxonomic composition of diarrheal case samples. For instance, samples from Nigeria and Mozambique showed a high abundance of *Klebsiella*, whereas *Bifidobacterium* was more prominent in Ethiopia and Tanzania. Interestingly, *Streptococcus* was highly abundant in samples from Tanzanian children with diarrhea and their siblings, more so than in any other country, potentially reflecting country-specific differences related to dietary, health-related practices, or environmental exposure patterns. Although the *Streptococcus* may not be the primary cause of the diarrhea in the children, its high presence could signal gut disruption[42,43]. To investigate whether these variations could provide useful biomarkers for population-level microbiome surveillance, further studies would have to be conducted.

Beyond static taxonomic profiles, notable temporal patterns were observed in the abundance of FBD-related taxa in the sewage samples. A clear increase in total read counts mapping to known FBD pathogens was observed in sewage samples collected in 2021, suggesting a potential rise in pathogen burden during that period. While the overall load of FBD-associated taxa increased, the relative composition remained consistent, indicating that the signal was not driven by a single expanding pathogenic lineage but by broader community-level changes. This temporal signal coincided with the COVID-19 pandemic, which may have altered healthcare access, sanitation practices, or other community-related practices, which in turn has impacted the abundance of FBDs in the period.

Hierarchical clustering of CLR-transformed abundances for the 50 most variable microbial species revealed limited overlap between microbial communities across different reservoirs, with samples tending to group by ecological source (e.g., Human, animal, environmental) rather than by country, which the PCA also confirmed. For instance, poultry and bovine samples consistently clustered apart from human and sewage samples, regardless of geographic origin. This suggests that reservoir-specific microbial signatures are maintained across settings, likely driven by host physiology, habitat, and microbial interactions rather than local environmental conditions. For example, the gut microbiota of poultry, regardless of their origin, may be dominated by similar genera due to shared gut physiology and diet across avian hosts[44,45]. The clear separation between reservoirs also implies that cross-reservoir transmission of foodborne pathogens may be sporadic or constrained, emphasizing the need for targeted surveillance strategies that consider reservoir ecology.

Reconstruction of high-quality MAGs for *E. coli* and *Campylobacter jejuni* enabled direct comparison between genomes obtained from the metagenomic samples and those recovered via traditional culturing and WGS. These MAGs revealed diverse STs, few of which were redundant with STs observed in the isolate dataset. Despite differences in ST assignment, the MAGs integrated phylogenetically with the isolate dataset. SNP-based phylogenetic analysis showed that several *E. coli* MAGs clustered within clades composed of isolates from similar sources, such as children with diarrhea or bovine/cattle samples. A single MAG with an unassigned ST grouped closely with a well-supported clade of ST131 isolates, suggesting phylogenetic relatedness despite the absence of a clear ST profile.

Pairwise comparisons of ANI between MAGs and isolates revealed similarity values (range: 95.75–96.9%), confirming that the MAGs represent the same species as the isolate genomes. However, none of the MAGs reached ANI thresholds (>99.0%) typically associated with clonal or strain-level identity, indicating that while the MAGs captured

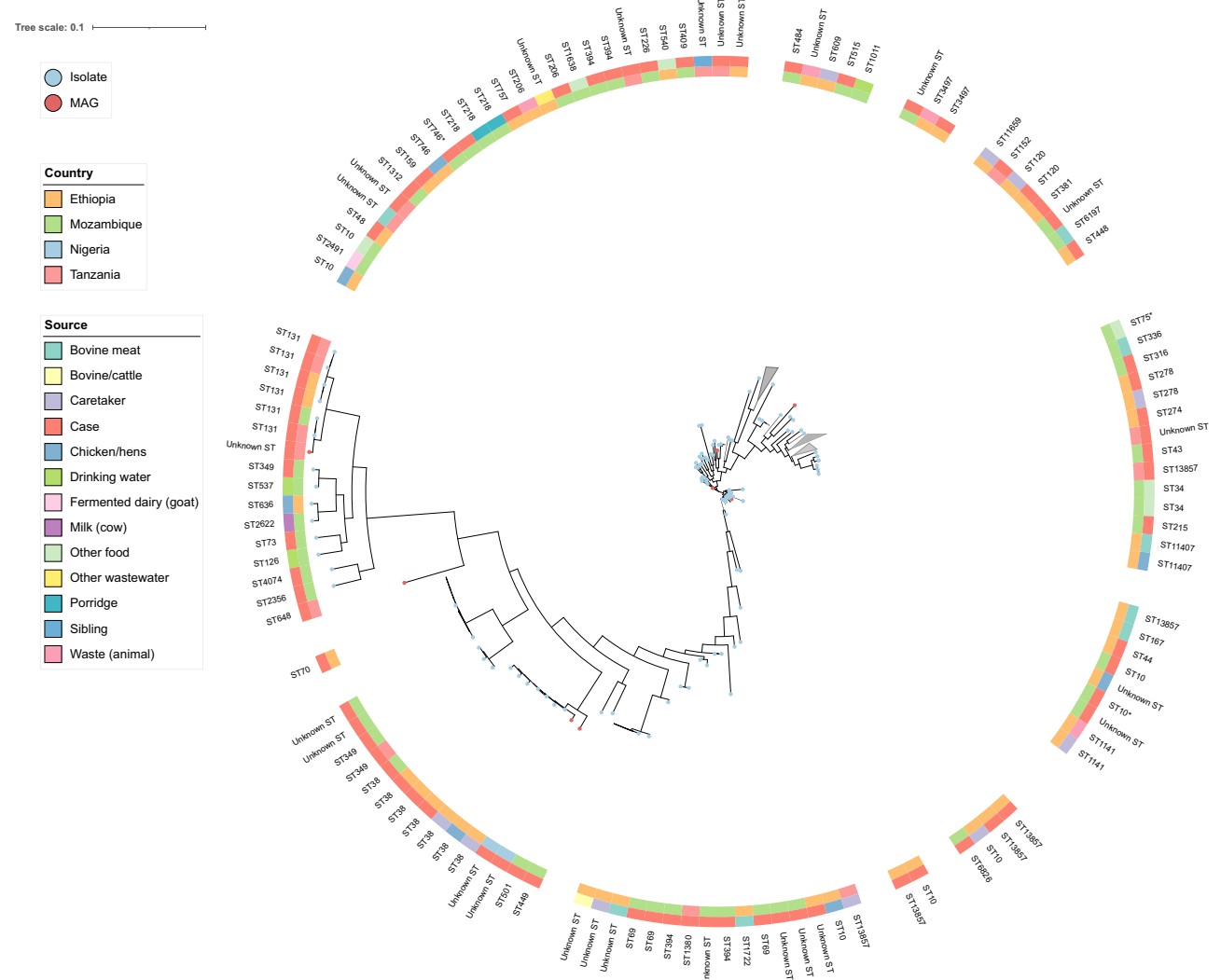

**Fig. 5 | Single-nucleotide polymorphism (SNP)-based phylogeny of whole-genome sequence (WGS).** *E. coli* isolates and metagenome-assembled genomes (MAGs) across four African low- and middle-income countries. Maximum-likelihood phylogenetic tree constructed using IQ-TREE2 from a SNP alignment generated by CSIPhylogeny (*n* = 204 genomes). The inner ring indicates country of origin, and the outer ring denotes sample source (e.g., human, animal, environmental). Sequence types (STs) are labeled alongside the tips. Blue dots indicate *E. coli* isolates obtained through WGS, while red dots indicate high-quality MAGs.

species-level diversity, they likely represent distinct lineages not recovered by culturing.

As part of this study's aim to generate an overview of FBD-causing pathogens in a sub-Saharan setting, these results demonstrate the feasibility of using recovered MAGs of foodborne pathogens from complex, culture-independent samples as a complementary tool in FBD surveillance. Our findings show that genomic information of epidemiological relevance, including MLST, phylogenetic placement, and virulence gene content, can be extracted from MAGs. Furthermore, the co-clustering of MAGs and WGS-derived isolates from the same or similar sources in phylogenetic analyses highlights the utility of combining both approaches to trace microbial signatures across different reservoirs. This is an important element because it demonstrates that metagenomic sequencing can recover genomic patterns that are consistent with isolate-based WGS, even in the absence of direct sample pairing, supporting its use as a complementary tool for pathogen surveillance, particularly in settings where culturing is infeasible or incomplete.

However, the use of MAGs for high-resolution epidemiological surveillance and applications, such as outbreak detection, is still limited by key technical constraints. MAGs must reach stringent quality

standards to enable reliable inferences at the strain level[46]. Achieving such quality is heavily dependent on the relative abundance of the target organism in each sample, which can be difficult to obtain for pathogens that are not abundantly present in the samples collected. In this study, we were only able to recover MAGs for *E. coli* and *C. jejuni* from a subset of the samples, reflecting the fact that pathogens of interest must be both present and sufficiently abundant to be effectively assembled and binned. Nevertheless, other studies have demonstrated that metagenomic sequencing can be used to identify and characterize bacterial strains during outbreaks without laboratory culturing, particularly when high-quality MAGs can be reconstructed or when strain-resolved analysis is feasible through deep sequencing or hybrid assembly[19,47,48].

Even when high- or medium-quality MAGs cannot be recovered due to low pathogen abundance, high community complexity, or insufficient sequencing depth, read-level classification tools such as KMA and Kraken2 offer powerful alternatives for surveillance purposes. The direct mapping of metagenomic reads to reference databases allows for species- or genus-level abundance without the need for genome assembly. In this study, such approaches were key in quantifying the presence of FBD-causing pathogens across complex

sample types like sewage or livestock samples, where full genome reconstruction was not always possible. Provided that sequencing depth is sufficient, abundance estimates derived from read mapping can be used to track temporal trends, identify geographic hotspots, or detect unexpected shifts in pathogen prevalence, as shown in other studies[21,49].

Despite the promise of metagenomic approaches, several important limitations were also evident. One key challenge was the detection of antimicrobial resistance genes in MAGs, which were sparsely detected. This may be attributable to the fact that many AMR determinants are located on mobile genetic elements such as plasmids, which are often not accurately assembled or binned during metagenomic reconstruction[50]. The fragmentation of short-read assemblies and the difficulty in capturing horizontally transferred elements can make the detection of these clinically relevant features difficult in MAGs, and as such, MAGs might not be the best use for surveillance of these elements[51,52]. Furthermore, the interpretation of metagenomic trends and source–reservoir relationships in this study is constrained by the limited number of samples from some reservoirs, which may have reduced our ability to capture the full ecological and genomic diversity of these FBD-causing pathogens. Finally, a consistent barrier to both isolate-based and metagenomic surveillance in Africa is the lack of representative genomic data in public databases. This was reflected in our finding that several MAGs and isolates could not be assigned known sequence types or predicted serovars, suggesting that lineages circulating in African settings are underrepresented in current reference frameworks. This limitation not only affects typing accuracy but also restricts broader epidemiological interpretation. Consequently, there is a pressing need to expand genomic datasets from LMICs, both to improve pathogen detection and to inform global comparative studies.

An additional limitation of this work was that there was no overlap between samples submitted for WGS and metagenomic sequencing. This precluded direct comparisons between isolate-level and community-level data. Consequently, potential ecological links between cultured isolates and their surrounding microbial communities could not be fully resolved. Future studies incorporating paired WGS and metagenomic analyses, e.g., by using PCR screening as a first step, would provide stronger resolution of pathogen dynamics and provide more direct evidence for the complementary value of integrating both approaches in surveillance systems.

While WGS remains a powerful tool for high-resolution pathogen tracking, it is restricted to culturable organisms and often resource-intensive. In contrast, metagenomics enables simultaneous detection of multiple pathogens in complex samples and has been successfully applied in surveillance of zoonoses and foodborne diseases in other settings[19,21,53]. Integrating read-based approaches with WGS can lead to a robust, complementary framework for LMICs, where resource constraints and infrastructure variability require different surveillance strategies. As demonstrated in this study, leveraging both WGS and metagenomic data can enhance the detection, characterization, and contextual understanding of FBD pathogens, laying the fundamentals for cross-sectoral surveillance systems tailored to the needs of African LMICs.

## Methods
### Sample collection
Data were collected in healthcare facility-based quantitative and qualitative cross-sectional studies conducted in the four African countries of interest (Ethiopia, Mozambique, Nigeria, and Tanzania) between 2019 and 2023. A schematic overview of the study design, including sampling strategy, sequencing methods, and subsequent analyses for both whole-genome sequencing (WGS) and metagenomic data is shown in Fig. 6. Stool samples were collected in collaboration with pediatricians and/or clinicians at the local health-care facilities in the study areas, where diarrheic children below 5 years of age seeking medical treatment were enrolled in the study. Participants included children across the age range of 0–5 years, their caregivers, and, when available, siblings. Sex was not distinguished, as environmental exposures were the primary variables of interest. Caregivers provided written and oral informed consent for the participation of the child, themselves, and any siblings.

Participation in household visits was voluntary. To acknowledge the time and effort of families during household visits, participants who consented were given a small gift (~1 kg of rice, salt, or soap, valued at a maximum of 3 USD). This gift was provided solely to thank families for their participation in the household visit and was independent of any hospital services or medical care. Participants were informed that participation was voluntary and that refusal would not affect the care received.

If the caregivers consented to participate, fecal samples were obtained from both the child and its caregiver. If the caregivers consented, samples were then obtained from the child's recent exposure environment, which included fecal samples from the caregivers of the child or siblings, and samples from food-, water-, or animal sources. Food products, including meat samples, droppings from domestic animals, and wastewater, were also sampled from local abattoirs and marketplaces that were directly or indirectly linked to the households. Sewage samples were collected from selected urban and rural sampling sites that were representative of the population and varied depending on the country's population size in the study sites. Sampling sites included wastewater treatment inlets, septic tanks, or outlets, where the sewage ran directly into a body of water. Relevant epidemiological data were collected for each sample, including the GPS coordinates of the sampling location.

### Culture, isolation, and WGS
Samples were collected by trained personnel employed by the local universities or research centers during the household visits. All samples were collected in a sterile container and kept in a cold box containing ice packs before being transported to local research centers or laboratories on the day of collection. To detect microbial pathogens related to foodborne diseases, the samples were processed using a Standard Operating Procedure (SOP) created for the FOCAL-project for each target pathogen (Non-typhoid *Salmonella* spp. and *Shigella* spp., diarrheagenic *E. coli* spp., and *Campylobacter* spp.) (Supplementary Methods 1 and 2). Only Ethiopia examined samples for *Campylobacter* spp. Isolates obtained from culturing were stored at −80 °C before they were shipped to the sequencing facility. Isolates were shipped on ice for WGS at either Kilimanjaro Clinical Research Center (KCRI) (Moshi, Tanzania), University of Pretoria (UP) (Pretoria, South Africa), or Admera Health Biopharma Services (San Diego, USA). Different library preparation kits and sequencing instruments were used, depending on which laboratory handled the samples (Supplementary Data 1). Samples sequenced at KCRI were prepared using the Illumina DNA Preparation Kit (formerly known as Nextera DNA flex) and sequenced using an Illumina NextSeq 550 instrument with 2 × 150 bp paired-end reads.

Samples sequenced at UP were prepared using the MGIEasy Universal DNA Library Prep Kit and sequenced using an MGI DNBSEQ-G400 instrument with 2 × 150 bp paired-end reads.

Samples sequenced at Admera Health Biopharma Services were prepared using either a Nextera XT Library Preparation kit or a Tagmentation Library Preparation kit for Illumina and sequenced using an Illumina NovaseqX Plus 10B instrument with 2 × 150 bp paired-end reads.

### Metagenomic sequencing
A randomly selected subset of the samples collected from the four countries, along with sewage samples representing the populations

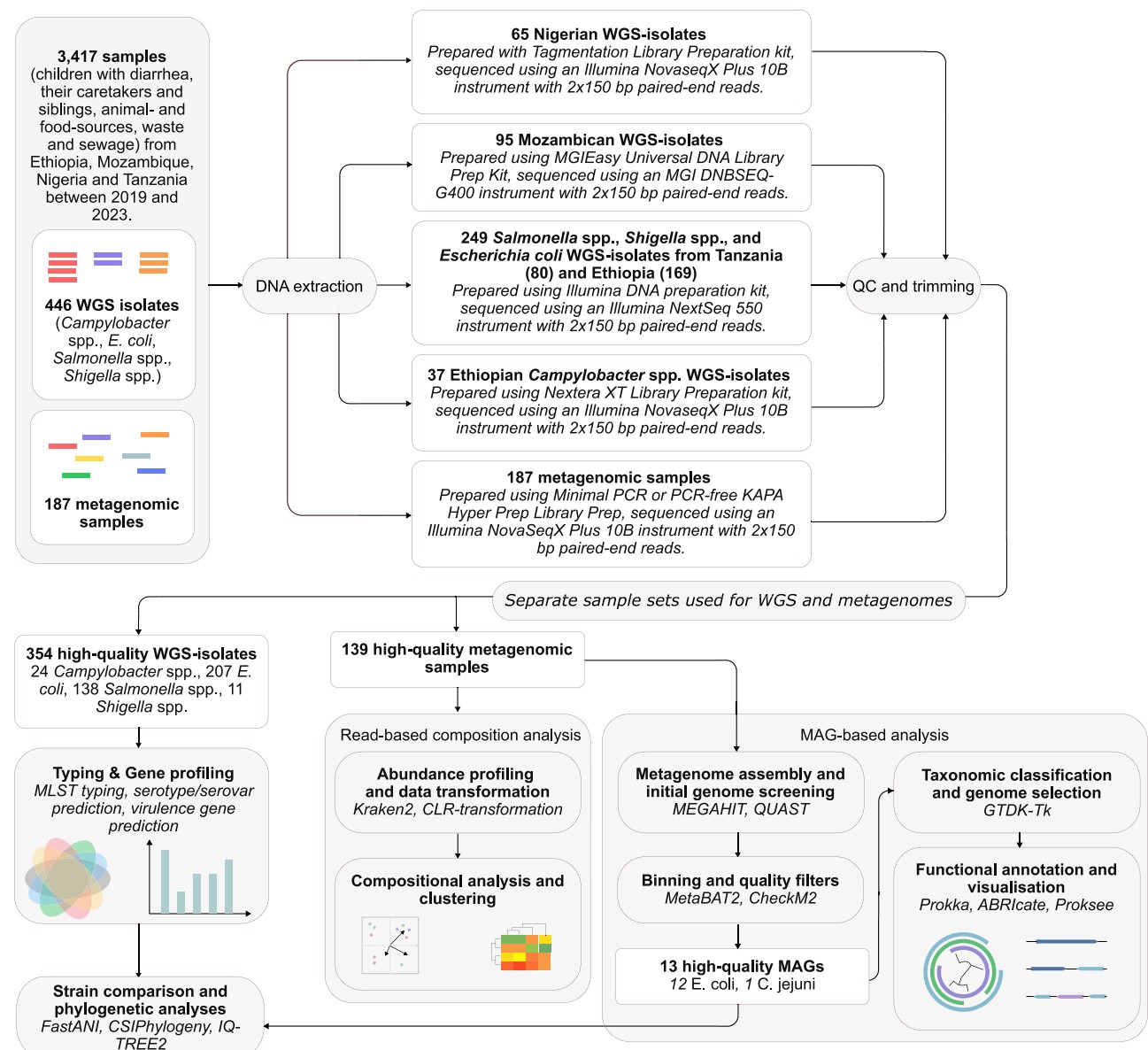

**Fig. 6 | Overview of sampling, sequencing, and analytical workflow for the whole-genome isolates (WGS) and metagenomic data.** Samples were collected from human, animal, food, water, and environmental sources across four African countries between 2019 and 2023. Importantly, the datasets were processed for WGS and metagenome sequencing separately, using different subsets of samples. For WGS, cultured isolates were subjected to DNA extraction, library preparation, and paired-end sequencing. Reads were quality-checked (QC), trimmed, assembled, and taxonomically classified, followed by multi-locus sequence (MLST) typing, serotyping, and functional annotation. Metagenomic DNA was extracted directly from raw samples, sequenced, and analyzed for community composition and taxonomic profiling. Reads were classified using Kraken2/Bracken and KMA, and relative abundance was estimated using centered log ratio (CLR) transformation. Metagenomes were assembled using MEGAHIT and screened for pathogens of interest using QUAST. MetaBAT2 was used to bin contigs into putative genomes, and CheckM2 was employed to evaluate their completeness and contamination, retaining only high-quality metagenome-assembled genomes (MAGs) for downstream analysis. These MAGs were classified taxonomically using the Genome Taxonomy Database Toolkit (GTDB-Tk) and compared to cultured isolates using average nucleotide identity (ANI) and phylogenetic reconstruction. Together, MAGs and isolates were analyzed to explore pathogen diversity, abundance, and ecological structures across samples.

in the study areas, were shipped to Admera Health Biopharma Services (San Diego, USA) for metagenomic sequencing. Samples for metagenomic sequencing were selected without knowing the results of culture, isolation, and WGS, resulting in very few samples undergoing both sequencing procedures. DNA extraction was performed using QIAampFast DNA Stool kits (Qiagen, Denmark), and library preparation was conducted using either minimal PCR or PCR-free KAPA Hyper Prep kit and sequenced using an Illumina NovaSeqX Plus 10B instrument with a targeted sequencing depth of minimum 85M paired-end reads per sample[54]. A comparison of the bacterial DNA

content in the samples using either the minimal PCR or PCR-free library preparation did not reveal any detectable difference in the genome composition for any of the samples (Supplementary Fig. 1, Supplementary Table 1).

## Quality control and assembly
Paired-end reads from both WGS isolates and metagenomic samples were initially assessed for quality using FastQC (v0.11.5)[55]. Adapter trimming and quality filtering were performed using BBDuk (v36.49) (part of the BBTools suite)[56] with a Phred score threshold of Q20 to

remove low-quality bases (corresponding to a 1% error rate) and eliminate common sequencing adapters.

WGS reads were assembled using SPAdes (v3.11.0)[57], using a multi-k-mer approach for assembly refinement ($k$ = 21, 33, 55, 77, 99, 127), strict coverage cutoffs to exclude low-confidence regions, and error correction to improve assembly accuracy. Quality of assembled genomes was evaluated using SeqKit stats (v2.4.0)[58] and manually inspected for species-specific expectations, including total assembly length, number of contigs, and N50 values. Assemblies with excessive fragmentation or genome size deviating markedly from reference expectations were excluded from downstream analyses.

Metagenomic reads were assembled using MEGAHIT (v1.2.9)[59] with default parameters and a minimum contig length of 1000 bp. Resulting assemblies were assessed using QUAST (v5.2.0)[60] to evaluate assembly statistics and screened for the presence of target FBD-causing pathogens *Escherichia coli*, *Salmonella* spp., *Shigella* spp., and *Campylobacter* spp., based on contig content and taxonomic assignment. Only samples with evidence of these species were selected for further processing, including metagenomic binning and genome reconstruction.

### Genomic characterization and taxonomic profiling of WGS isolates

Species identified as *Salmonella* spp., *E. coli*, and *Campylobacter* spp. were characterized using seven-loci Multilocus Sequence Typing (MLST), performed via the pipeline developed by the Center for Genomic Epidemiology (v2.0.9, database version 2023-06-19). Typing schemes were applied based on already established MLST definitions for *Salmonella* spp.[61], *E. coli*[62], and *Campylobacter* spp.[63].

Serotype prediction was conducted using SeqSero (version 1.1.1)[64] for *Salmonella* isolates and SeroTypeFinder (v2.0.0)[65] for *E. coli* isolates, using default parameters.

### Taxonomic profiling and abundance estimation of metagenomic reads

Metagenomic reads were aligned against a comprehensive GenBank-derived database (downloaded 2021-09-22) using KMA (v1.4.2)[66] to determine the bacterial composition of the samples. The database included complete and draft genomes from bacteria, archaea, protozoa, fungi, plasmids, and viruses. The abundance was calculated as the number of fragments aligned to each genome, adjusted for sequencing depths and number of fragments in the samples, using the centered log transformation (CLR). The CLR transformation is useful for quantifying abundance in compositional data because it maintains the same ratio from a sample with few fragments and a sample with many fragments[67]. The CLR transformation for a given sample $x$ with a given feature $D$ is given as

$$\text{clr}(x_i) = \ln\left(\frac{x_{1,i}}{g(x_i)}\right), \ldots, \ln\left(\frac{x_{D,i}}{g(x_i)}\right) \quad (1)$$

where $x_i$ is the *i*th sample and $g(x_i)$ is its geometric mean, which is calculated as $\sqrt[D]{x_1 \cdot x_2 \cdot \ldots \cdot x_{D,i}}$.

To visualize compositional trends in pathogen abundances, relative abundances and CLR-transformed abundances were calculated. A boxplot was constructed to show the distribution of CLR-transformed abundance values for each FBD-causing pathogen across all available sources. Relative abundances were approximated by applying an exponential transformation to the CLR-transformed values. For each pathogen, the CLR-transformed values were exponentiated (i.e., exp(CLR)) and normalized within each sample to sum to 100%.

As an additional task, metagenomic reads were also taxonomically classified using Kraken2 (v2.1.3)[68] with a standard Kraken2 database (downloaded 2023-07-23). Genus-level relative abundances were estimated using Bracken (v2.6.2). For visualization, only the 15 most

abundant genera across all samples were retained, and all remaining genera were aggregated under an "Other" category to simplify interpretation.

### Principal component analysis

To explore patterns in the microbial communities in the samples and identify taxa associated with the sample source, principal component analysis (PCA) was performed on the CLR-transformed abundance from all metagenomic samples. Arrows representing the top 10 contributing species were projected into the ordination space to improve interpretation. The direction of each arrow reflected the gradient of increasing abundance for the corresponding species, while the length of the arrow reflected its relative contribution to the observed variation. We applied a permutational multivariate analysis of variance (PERMANOVA) to test the significance of the differences in the microbial diversity across various isolation sources, seasons, and countries.

### Diversity analysis

Ecological and geographical patterns in pathogen composition were investigated using a clustered heatmap based on the CLR-transformed abundance values. The 50 most variable pathogens were selected by identifying species with a high standard deviation across all samples. Abundance values were aggregated by source and country, using the mean CLR-transformed value within each group. To ensure cross-country comparability, source categories represented by only a single country were excluded from the analysis.

Hierarchical clustering was performed separately on the rows (source–country combinations) and pathogens. Dendrograms were generated using Ward's linkage with Euclidean distance. The resulting heatmap displayed group-level CLR-transformed abundances, with rows clustered by ecological and geographical similarity, and columns clustered by co-occurrence structure across groups.

### Strain-level comparison between MAGs and isolates

Contigs ≥1500 bp were subjected to binning using MetaBAT2 (version 2.17)[69], and resulting metagenome-assembled genomes (MAGs) were evaluated for completeness and contamination using CheckM2 (version 1.0.2)[70]. Only MAGs meeting the criteria for medium or high quality defined as >90% completeness and <5% contamination for high quality, and >50% completeness and <10% contamination for medium quality—were retained for downstream analyses, in accordance with MIMAG guidelines[71].

To assess whether MAGs recovered from metagenomes represented the same strains as cultured isolates, we computed pairwise average nucleotide identity (ANI) using FastANI (v1.34.0)[72] between all isolate genomes and recovered MAGs. Pairs with >99.9% ANI were considered potential strain-level matches[72]. Phylogenetic relationships were inferred based on single-nucleotide polymorphism (SNP) differences using CSIPhylogeny (v1.4.0), a pipeline that detects and filters high-quality SNPs, performs site validation, and generates a concatenated SNP alignment for tree inference[73]. SNP calling was performed with a minimum SNP quality score of 30, minimum depth of 10, relative depth threshold of 10% (based on average depth), Z-score filter of 1.96, mapping quality filter of 25, and SNP pruning at a distance of 10 nucleotides. Maximum-likelihood phylogenetic inference was then performed using IQ-TREE2 (v2.4.1)[74]. The alignment of nucleotide sequences was analyzed with ModelFinder to[75] select the best-fit substitution model, and branch support was evaluated using 1000 ultrafast bootstrap replicates[76]. The resulting maximum-likelihood tree was visualized using iTOL (Interactive Tree of Life, v7.1.0).

### Functional annotation and virulence gene detection

Both MAGs and isolate genomes were annotated using Prokka (v1.14.5)[77] for gene prediction. Virulence genes were identified using

ABRicate (v1.0.1) with the VFDB[78] and CARD[79] databases (downloaded December 3, 2021). Selected high-quality *Campylobacter* spp. and *E. coli* MAGs were further visualized using Proksee[80] to depict circular genome structure and gene content.

## Ethics and inclusion statement
Our research project has been carefully designed and implemented to align with the principles outlined in the Cape Town Statement, which emphasizes the importance of equitable and ethical research practices in LMICs. The project was conducted as a multi-country collaborative partnership, involving local researchers employed at local universities or research institutions from Nigeria, Tanzania, Ethiopia, and Mozambique. The researchers played an integral role in every stage of the research process, from study design and planning to data collection and analysis. Local researchers were entrusted with data ownership and were encouraged to publish findings in local and international journals.

The relevance of the research was determined collaboratively with local partners, ensuring that the study focused on diseases and public health issues of local significance. The principal investigators from each participating country led the research efforts within their respective regions, engaging local data collectors, research areas, health workers, and community leads. They were also responsible for involving local stakeholders and governmental representatives as needed. This collaborative approach extended to capacity-building initiatives, such as genomic data and epidemiological workshops, where researchers from all participating universities were trained in bioinformatics analysis and epidemiological modeling.

To maintain data consistency across the project, all partner universities followed standardized protocols for culturing and isolation. While we aimed to conduct sequencing within Africa, primarily in Tanzania and South Africa, technical challenges necessitated sending some samples to the USA for sequencing. Despite this, efforts were made to involve local researchers in the sequencing process to enhance their expertise. Ethical considerations were rigorously adhered to, with all studies receiving approval from the respective countries' ethics committees.

Our research involved the non-invasive collection of fecal samples, minimizing any potential harm to animals. Given that animal welfare concerns were negligible, we maintained high ethical standards in line with the principles of non-invasive and environmentally responsible research. The project ensured that participants remained anonymous, with data collected and communicated in local languages by health workers familiar with the communities. This approach minimized risks of stigmatization or discrimination.

Regarding risk management, standard safety protocols were followed in the collection, transport, and laboratory handling of biological samples, such as feces and sewage. We ensured that the research was conducted with minimal risk to both participants and researchers. While the project aimed to perform most research within the countries of origin, limited sequencing capacity necessitated the transfer of some samples abroad. However, all data were promptly shared with the participating countries, and local researchers were involved in the sequencing process whenever possible.

In preparation for this project, significant effort was made to review and cite relevant local research. Throughout the research and manuscript preparation, we prioritized citing locally relevant studies, contributing to the body of knowledge within the respective countries, and ensuring that the project's outputs were both scientifically robust and locally meaningful.

## Ethics declarations
The study was conducted in accordance with the Declaration of Helsinki. The study protocols were reviewed and approved by the appropriate Ethics Committee in the respective participating countries. In Ethiopia, the study obtained ethical clearance from the Institutional Health Research Ethics Review Committee (IHRERC) of Haramaya University and the National Research Ethics Committee with a letter written on February 11, 2020 (Ref. Number: MoSHE/RD/14.1/9849/20). In Mozambique, the study obtained ethical clearance from the Institutional Committee on Bioethics in Health of the Faculty of Medicine/Central Hospital of Maputo with a letter written on December 12, 2019 (Ref. Number: CIBS HM&HCM/092/2019). In Nigeria, the study obtained ethical clearance from the Federal Medical Centre, Abeokuta Health Research Ethics Committee (HREC) with a letter written on November 27, 2019 (Ref. Number: NHREC/08/10-2015). In Tanzania, the study obtained ethical clearance from the Kilimanjaro Christian Medical University Research Ethics Committee (CRERC) and the National Institute of Medical Research (NIMR) (Ref. Number: NIMR/HQ/R.8a/Vol. IX/3273) with a letter written on November 27, 2019 (Ref. Number: 2496). The aims and study objectives were thoroughly explained to the parents of the diarrheal child in their local language(s). Each parent decided whether to consent to his or her child's participation in the study. The parent(s) were also informed of their freedom to withdraw from the study at any time without consequences. In addition, written informed consents were obtained from the child's parent to publish this manuscript. Study identification numbers were assigned to all participants, and access to data was restricted to researchers to ensure confidentiality.

## Reporting summary
Further information on research design is available in the Nature Portfolio Reporting Summary linked to this article.

## Data availability
The data that support the findings of this study are available in the European Nucleotide Archive (ENA) under accession number PRJEB73590. This includes both metagenomic sequencing data and whole-genome sequences of cultured isolates. Individual accession numbers for the metagenome-assembled genomes (MAGs) and assembled isolate genomes (FASTA files) are provided in Supplementary Data 1. All data are publicly available without restriction and can be accessed directly through the ENA portal using the accession number above. Processed data used to generate Figs. 1 and 3 are available in the accompanying Source Data files. Source data are provided with this paper.

## Code availability
All analyses were performed using publicly available software packages and tools, as described in the "Methods" section. The only scripts used in this study were simple bash scripts for running these tools and scripts for plotting or visualizing processed results. No custom analytical pipelines or data processing functions were developed that would impact the reproducibility of the analyses.

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

## Acknowledgements

This study is part of the "Foodborne disease epidemiology, surveillance, and Control in African LMIC (FOCAL)" Project, a multi-partner, multi-study research grant co-funded by the Bill & Melinda Gates Foundation and the Foreign, Commonwealth & Development Office (FCDO) of the United Kingdom Government (Grant Agreement Investment ID OPP1195617). The funders of the study had no role in study design, data collection, data analysis, data interpretation, or writing of the report. We

acknowledge all researchers and stakeholders involved in the project, thanking them for their valuable contributions. We thank Saria Otani for her help with the organization of the wetlab. We would also like to thank Thomas Nordahl Pedersen and Hannah-Marie Martiny, Binyam Desta, and Francisco Avila for feedback on the manuscript.

## Author contributions

C.T. and T.H. conceptualized the paper. Formal analysis and writing of the manuscript were done by C.T., with help from D.B., E.A.H., T.S.O., J.G., and S.F. T.H. supervised the process. Funding was acquired by T.H., T.G., S.M.P., E.M.S., B.T.M., C.A., O.E.F., B.M., and E.M.B. Revision was led by C.T., with input from all authors, including H.K., G.A., E.M.B., K.M.T., S.M.P., P.M.K.N., S.F., and T.H.

## Competing interests

The authors declare no competing interests.

## Additional information

[1]National Food Institute, Technical University of Denmark, Kgs. Lyngby, Denmark. [2]School of Environmental Health Science, College of Health and Medical Sciences, Haramaya University, Dire Dawa, Ethiopia. [3]Faculty of Sciences and Veterinary Faculty, Eduardo Mondlane University, Maputo, Mozambique. [4]Department of Biological Sciences, Mountain Top University, Ibafo, Ogun State, Nigeria. [5]Biotechnology Laboratory, Kilimanjaro Clinical Research Institute, Moshi, United Republic of Tanzania. [6]Department of Microbiology, KCMC University, Moshi, United Republic of Tanzania. [7]Department of Consumer and Food Sciences, University of Pretoria, Pretoria, South Africa. [8]College of Veterinary Medicine, Haramaya University, Dire Dawa, Ethiopia. [9]School of Public Health, KCMC University, Moshi, United Republic of Tanzania. [10]Department of Scientific Research, Instituto Superior de Ciências de Saúde, Maputo, Mozambique. [11]Department of Microbiology, University of Lagos, Lagos, Nigeria. [12]Centre for International Health, University of Otago, Dunedin, New Zealand. ✉e-mail: ceth@dtu.dk

