## [Transparent Peer Review file · Nature Communications]

Using Metagenomics and Whole-Genome Sequencing to characterize enteric pathogens across various sources in Africa

Corresponding Author: Ms Cecilie Thystrup

Version 0:

Reviewer comments:

Reviewer #1

(Remarks to the Author)

The study titled “Integrative Metagenomics and Whole Genome Sequencing Analysis Across Multiple Sources in Four Low- and Middle-Income African Countries: the FOCAL-study” authored by Thystrup et al with control number NCOMMS-24-62410 describes the examination of isolates and metagenomic data from low and middle income countries with a focus on food borne disease pathogens in these settings. The authors examined isolates and metagenomic samples from four locations and a number of sources including humans, wastewater and livestock animals. The number of species that were examined across all of the different locations and sample types results in relatively low numbers in essentially all study groups and species, preventing the application of robust statistical methods on the presence or absence of traditional markers of virulence such as serotype or sequence type. The inclusion of metagenomic studies and data is novel and innovative; however, the authors miss a chance to link the isolates obtained with the metagenomic data and take only a superficial examination of this data. This is an interesting study with unrealized potential. Overall, the study lacks rigor due to the low numbers of isolates or metagenomic samples in any group, as well as a number of fundamental issues with the release of the associated data.

Major Comments:

The authors have not provided an inventory of which isolates were sequenced from which locations. It may be possible to work backwards from the ENA/pubic data to identify the isolates, but burden should not be on the reader. This is easily included as a Supplemental table or dataset. Additionally the public data is labelled as follows: “WGS and Metagenomic samples from Ethiopia, collected as part of the FOCAL - Foodborne disease epidemiology, surveillance, and Control in African LMIC” - it is unclear if only the data/isolates from Ethiopia is public. If this is the case, then the study should not move any further forward until ALL the data is public. This is a requirement of this journal and genomic investigators in general. Leaving this requirement in the hands of the authors is not acceptable and this data needs to be accessible before this manuscript moves any further forward. This is a fundamental component of these types of studies.

The public data is only the primary read data, whereas the majority of the analysis is completed on assembled datasets, yet these assemblies are not public. There are many ways in which assembly can go wrong. The authors cite an “in house QC- and assembly pipeline” – again which is not public, thus making it impossible to verify, validate or evaluate. This is also a fatal flaw of this study.

There is a general lack of statistical tests associated with most/many of the conclusions throughout the manuscript. This is a significant flaw of the study and needs to be addressed through including the appropriate statistical tests or explaining why these tests cannot or should not be included. When statistics are included they are not well explained nor are they used in a way that is the standard in the field.

It is unclear why the metagenomic reads were not mapped against the isolate sequenced from which they were derived? This would be an interesting aspect of the study to determine if the cultured isolate is abundant in the metagenomic data and if that could be correlated to the clinical syndromes (diarrhea) observed or the origin of the samples. This would have also allowed the authors to support some of the conclusions that are currently unsupported in the discussion regarding the

stability and/or distribution of different pathogens.

The discussion of the metagenomic data and the implications for health and disease is limited and not well supported, as this type of approach is not often undertaken.

Minor Comments:

The lack of consistent or abundant serotypes of *E. coli* or species in the case of *Campylobacter* is not surprising considering the relatively low number of isolates of each species from each of the geographic locations. While the authors want to make a scientific point of this there is no statistical power to these studies and this can be minimized or removed from the manuscript.

Figure 1 is so small that it is currently unreadable without significant expansion on the screen. It should be rotated to landscape and expanded. The same can be said for many of the figures.

Reviewer #2

(Remarks to the Author)

In this study, Thystrup et al. report some really valuable and interesting findings about the prevalence of food-borne pathogens across four African countries, investigating the prevalence across different hosts and environments using both whole genome sequencing and shotgun metagenomics. As studies and data collected in low- and middle-income countries are scarce, the data produced would be useful for future research and integration in surveillance databases. I applaud the choices done by the research team by performing, when possible, the research in the countries in which each part of the study was conducted, and the fact that the data obtained by their analyses was reported back to the origin. I have some questions and remarks on both the methodology used to conduct the study and on the results obtained.

The authors stated in the methods section that the main subjects of study were children aged between 0 and 5 that were seeking medical treatment for diarrhea. Have the authors observed any changes in the composition of the microbiome between different age differences or differences between how severe the diarrhea was or the duration of the diarrhea? Have you found any differences in serotype prevalence and microbiome composition between individuals living in an urban or rural setting?

Also, I know that maybe the sample size does not allow you to do this type of analysis, but if both the children and the caretaker or sibling were exposed to the same source of FBP, and only the children is experiencing diarrhea, can you identify the same serotype in its caretaker/sibling?

Regarding the methodologies used, the collected samples were processed, prepared and sequenced using different kits and technologies, depending on the local availability, and as the authors discuss later in the paper, this could be a source of data inconsistencies. However, in the analyses this does not seem to be addressed since I do not see any batch effect correction. Have you looked at whether the different combinations of methods used yielded different quality of data, or whether the kit used were biased towards certain taxa?

It is stated that WGS isolates were assembled using SPAdes, but there is no mention whether the assembled genomes underwent a successive quality control to assess how good the assemblies were. Did the authors try to perform metagenomic assembly on the 158 metagenomic samples or tried to identify virulence factors from either the raw reads of MAGs?

The authors report the prevalence of the WGS isolates, but there is no supplementary table describing the species identified in the metagenomes.

What type of databases were used with KMA to perform the characterization of the reads? I assume from the description that genbank or refseq was used, but I do not understand why the plural form of database was used. In addition, were the WGS isolates characterized using the assembled genomes or the raw reads?

Figure 1 highlights the prevalence of some pathogenic species across the different types of samples, from the y-axis I see that the abundances sums up to 100%, but the text mentions that e.g. *E. coli* abundance is 4.14, so I assume that the relative abundances were re-normalized when plotted in figure 2, correct? If so, I would mention this particular in the caption since the figure could be a little misleading, or I would just plot the original values and collapse all the other taxa under an "Other" category.

How the changes in abundances across time and between countries were assessed? There is no statistical test reported in the text, and Figure 2 shows only the average between the countries. Also, what do you think is the reason behind the changes in average abundances in 2021? Could COVID infection be a possible reason?

The authors compared the alpha diversity between samples using a subset of species to find differences between countries and sample sources, and for doing that the Kruskal Wallis test was used. The method says that a post-hoc test was used as well, but the pairwise comparisons were not reported in the text.

What the ellipses' colors indicate in the ordination plots? Figure 5 has multiple ellipses with the same color. Are they indicating the 95% CI? Has it been tested for differences in composition between dry season and rainy season samples and how intragroup dispersion varies between the seasons?

Minor comments:

L101 has a capitalized species name, L453,454,465 *Salmonella* is not italicized
The citation for vegan is missing
Legend colors are not consistent across the figures

Figure 3 heatmap bottom annotation is missing the labels

Reviewer #3

(Remarks to the Author)

The authors present a collection of 625 WGS isolates and 168 metagenomic samples longitudinally collected from Ethiopia, Mozambique, Nigeria and Tanzania, with the goal of profiling the diversity, composition and genetic context of foodborne pathogens.

The study of foodborne pathogens in African countries is of relevance, and this project is well powered and spanned across several countries. The text is clear. However, while the proposed collection is very relevant for the field, most of the presented results are descriptive, with moderate indications of transmission events. In addition, there are widespread issues with the figures. Overall, in the current version, the figures and overall findings presented in this study are not up to the standards of the journal.

Major comments:

- The numbers of the figures are all wrong, and they do not coincide with the ones in the main text. The figures often contain formatting issues, are of difficult interpretation and plots are not aligned within the panel. In particular:
 - o Figure 1 (or 4): The plots are not aligned, the bars of different width makes the plots difficult to compare across reservoirs. I recommend to modify this figure into a single horizontal line strip, with all barplots side-by-side, and metadata on top to indicate country of sampling and reservoir type. The authors can also choose to improve the figure in alternative ways, as long as the issues mentioned above are addressed. In addition, as this is the first figure of the manuscript, it should first provide an overview of the study design: displaying the total number of samples collected (for isolation and for metagenomic sequencing), their numbers for each country and each year, the reservoir investigated and the technologies used to do so.
 - o Figure 2 (or 5): As the study sampled at best n=4 African countries, the maps showing the whole continent are predominantly grey. This plot could be a barplot, with each bar representing a country. A barplot would allow to better distinguish the values, and better compare the values from the same country longitudinally. This figure could be included with the previous one.
 - o Figure 3 (or 6): Species names should always be italicized, clustering of rows and columns can be adjusted to occupy less space, leaving more space for the heatmap and improving readability. Metadata strips should be bigger.
 - o Figure 4 (or 7): Text on top of the subplots has formatting issues and extra spaces, subplots are not aligned. Boxplots of different widths are difficult to compare. Similarly to what suggested for Fig 1, and what done by the authors in Supplementary Figure S5, this figure would be greatly improved if the boxplots would be showed all in one line, next to each other and with equal widths. Legend is missing. It might help to add jitter data to better convey the data point distribution within the boxplots; outlier values should not be displayed (rhombuses).
 - o Figure 5 (or 8): The use of shaded areas covers the position and color of the underlying data points. Shaded areas should be substituted by transparent areas with a colored outline (and only the top 4-5 isolation sources to avoid cluttering).
 - o Figure 6 (or 9): similar comments to the previous figure: shaded areas on the top PCA make it hard to see the data points. Also, size of the data points should be increased.
- Figure 9 and table 5 are about seasonality, yet seasonality is not mentioned in the Discussions, nor the in the Methods. How are the dry vs wet seasons defined? Is there variability between the sampled countries? In addition, date of collection and associated seasonality are not reported as metadata in the Project ID on ENA.
- The analyses presented in the study are all quite high-level, with no in-depth dive into the genomic content of the isolated genomes. One way the authors could provide more functional context to this work is by comparing their foodborne pathogen isolated genomes with public sequences, identify mutations, structural rearrangements, carriage of antimicrobial resistance genes, reconstruct their functional potential. Depending on the sequencing depth (if above >5M reads after preprocessing), strain-level analysis could be performed on the metagenomic samples, which in this case would be extremely relevant as strains belonging to the same species can range from commensal to pathogenic. Strain level-information could inform if there are multiple strains of the same species present in the metagenomic samples, providing information on their competition within the host.

Minor comments:

- Line 98: I would recommend citing a landmark paper in the use of metagenomics to track out outbreaks: Loman NJ, Constantinidou C, Christner M, Rohde H, Chan JZ, Quick J, Weir JC, Quince C, Smith GP, Betley JR, Aepfelbacher M, Pallen MJ. A culture-independent sequence-based metagenomics approach to the investigation of an outbreak of Shiga-toxigenic *Escherichia coli* O104:H4. *JAMA*. 2013 Apr 10;309(14):1502-10. doi: 10.1001/jama.2013.3231. PMID: 23571589.
- Legend of figure 1 (or 4): it should specify "foodborne pathogen"
- Legend of Figure 2 (or 6): the metadata strips should also include information on the reservoir type (human, animal, waste), similarly to what done for Figure 1; "CLR-value" should be extended to changed to CLR transformed abundance.
- Legend of Figure 5 (or 8): the metadata strips should also include information on the reservoir type (human, animal, waste), similarly to what done for Figure 1;
- Legend of figure 5 (or 8): it should specify "Isolation source", instead of "iso_source".
- Supplementary Table S6 is too big to be provided as .doc document; it should be provided as .csv or .tsv, or excel.
- Targeted sequencing depth should be included in the methods sections.
- For metagenomic samples, the number of microbial reads after pre-processing should be included in the methods section.

Version 1:

Reviewer comments:

Reviewer #2

(Remarks to the Author)

The authors have addressed all the raised points, I have no further comments.

Reviewer #3

(Remarks to the Author)

The authors have addressed all of my previous comments, including the addition of the suggested analyses and the inclusion of more biologically relevant results. My only remaining suggestion is to rotate Figure 3 so that the taxa names appear as rows instead of columns, which would improve readability.

Reviewer #4

(Remarks to the Author)

As the mediator stepping in for Reviewer #1, my remit is to assess whether the authors have satisfied that reviewer's main concerns. The ancillary matter of data availability is largely resolved, although the genome assemblies themselves have not yet been released.

The second—and far more substantial—issue, which I share, concerns the claimed integration of metagenomic and isolate WGS data. The manuscript is still framed as an “integrative” analysis in which metagenomic sequencing corroborates or augments isolate-based whole-genome surveillance, yet the authors now acknowledge that the metagenomic and WGS datasets were generated from entirely different specimens. Without paired samples, the revised analyses amount to incidental overlaps: the metagenomic data show no strain-level concordance, uncover no biologically meaningful findings (e.g., novel lineages or strain carriage patterns), and yield no epidemiologically useful insight. In short, the metagenomic component can no longer sustain the study's unique selling point, leaving the central claim unsupported.

Rectifying this gap would require either generating a new, paired specimen set—clearly beyond the scope of a revision—or removing the metagenomic half of the study and rewriting the narrative accordingly, which would markedly dilute the manuscript's claimed novelty and impact. Given these constraints, Reviewer #1's main concern remains unresolved.

	Reviewer comment	Response
	Reviewer #1	
1	The study titled “Integrative Metagenomics and Whole Genome Sequencing Analysis Across Multiple Sources in Four Low- and Middle-Income African Countries: the FOCAL-study” authored by Thystrup et al with control number NCOMMS-24-62410 describes the examination of isolates and metagenomic data from low and middle income countries with a focus on food borne disease pathogens in these settings. The authors examined isolates and metagenomic samples from four locations and a number of sources including humans, wastewater and livestock animals. The number of species that were examined across all of the different locations and sample types results in relatively low numbers in essentially all study groups and species, preventing the application of robust statistical methods on the presence or absence of traditional markers of virulence such as serotype or sequence type. The inclusion of metagenomic studies and data is novel and innovative; however, the authors miss a chance to link the isolates obtained with the metagenomic data and take only a superficial examination of this data. This is an interesting study with unrealized potential. Overall, the study lacks rigor due to the low numbers of isolates or metagenomic samples in any group, as well as a number of fundamental issues with the release of the associated data.	We thank the reviewer for recognizing the novelty of integrating metagenomic data alongside isolate sequencing in the context of foodborne disease surveillance in Africa. We agree that sample size is a limitation in this type of study; however, metagenomic sequencing remains costly and logistically challenging in many low-resource settings, particularly when aiming for sufficient sequencing depth to recover high-quality data for analysis. This study was designed as an exploratory effort to evaluate the feasibility of using metagenomics for surveillance in these contexts, something that, to our knowledge, has not previously been done on this scale across human, animal, and environmental samples in sub-Saharan Africa. We also appreciate the reviewer’s point about the potential for linking isolate and metagenomic data. While it was not possible to match metagenomic samples directly to their corresponding isolates, we went back and strengthened the manuscript by incorporating additional comparative analyses between high-quality MAGs and isolate genomes. We have restructured and expanded key sections of the manuscript to clarify the analytical steps taken, and to ensure the study meets appropriate standards, even within the constraints of sample size. We hope these substantial additions and revisions help demonstrate the value of the dataset and the robustness of the analyses.

2	The authors have not provided an inventory of which isolates were sequenced from which locations. It may be possible to work backwards from the ENA/pubic data to identify the isolates, but burden should not be on the reader. This is easily included as a Supplemental table or dataset. Additionally the public data is labelled as follows: “WGS and Metagenomic samples from Ethiopia, collected as part of the FOCAL - Foodborne disease epidemiology, surveillance, and Control in African LMIC” - it is unclear if only the data/isolates from Ethiopia is public. If this is the case, then the study should not move any further forward until ALL the data is public. This is a requirement of this journal and genomic investigators in general. Leaving this requirement in the hands of the authors is not acceptable and this data needs to be accessible before this manuscript moves any further forward. This is a fundamental component of these types of studies.	In the revised manuscript, we have provided a comprehensive list detailing the sequencing technology and the library preparation kits used for each of the samples. The table will also include information on the serotypes, ST and the species identified, as well as the ENA IDs. This has been done for both the metagenomic samples and the single isolates (Supplementary Table, supplied as an .xlsx file. This should clarify any uncertainties in the data. For the labelling of the public data, that is a mistake on our part and we have changed the description of the public data repository to “Ethiopia, Tanzania, Mozambique and Nigeria” to accurately reflect that all WGS and metagenomic samples from the project have been uploaded to the same repository.
3	The public data is only the primary read data, whereas the majority of the analysis is completed on assembled datasets, yet these assemblies are not public. There are many ways in which assembly can go wrong. The authors cite an “in house QC- and assembly pipeline” – again which is not public, thus making it impossible to verify, validate or evaluate. This is also a fatal flaw of this study.	Most of the initial analysis of the whole genome isolates were run on genomes that were not assembled, so for that reason, we did not think to upload the assembled genomes as well. Regarding the ‘in-house QC and assembly pipeline,’ we realize our wording in the methodology was unclear. It refers to a simple Bash script that runs FastQC for quality assessment, bbduk for adapter removal and quality trimming, followed by another FastQC run with results summarized in MultiQC. Describing it as an ‘in-house script’ may have been misleading, as it only automates the use of publicly available tools. This has been clarified in the revised manuscript.
4	There is a general lack of statistical tests associated with most/many of the conclusions throughout the	In the revised version, we went back and made sure to apply the appropriate tests where possible, and we have not clearly explained

	manuscript. This is a significant flaw of the study and needs to be addressed through including the appropriate statistical tests or explaining why these tests cannot or should not be included. When statistics are included they are not well explained nor are they used in a way that is the standard in the field.	what tests we used and why. In cases where we didn't use statistical tests, more explanation has been added.
5	It is unclear why the metagenomic reads were not mapped against the isolate sequenced from which they were derived? This would be an interesting aspect of the study to determine if the cultured isolate is abundant in the metagenomic data and if that could be correlated to the clinical syndromes (diarrhea) observed or the origin of the samples. This would have also allowed the authors to support some of the conclusions that are currently unsupported in the discussion regarding the stability and/or distribution of different pathogens.	We agree that this would have been a highly valuable analysis, and it was originally intended as part of the project. Unfortunately, we later realized that we had not sequenced the same samples for both metagenomics and WGS, and therefore could not directly trace isolate genomes back to their corresponding metagenomic reads. This was a limitation in the sampling design, and one we have learned from. That said, from a surveillance perspective, the ability to detect similar genomes across both metagenomic and WGS data still provides useful insight. In the revised manuscript, we have expanded the analysis to explore whether isolate-like genomes could be recovered in metagenomic samples, using phylogenetic comparisons between MAGs and WGS-derived isolates. This allowed us to evaluate the feasibility of using metagenomics as a surveillance tool, even in the absence of direct sample matching. We hope these additional analyses provide a more rigorous and biologically relevant perspective on the utility of metagenomics in genomic surveillance, particularly in resource-limited settings like sub-Saharan Africa.
6	The discussion of the metagenomic data and the implications for health and disease is limited and not well supported, as this type of approach is not often undertaken.	The discussion has been expanded to better highlight the potential of using metagenomics for surveillance, including its ability to detect pathogen signatures, track abundance over time and complement isolate-based surveillance using the results from our study. We have also added context from other studies to better support our interpretation and explain how this approach could be applied in future public health efforts.

7	The lack of consistent or abundant serotypes of E. coli or species in the case of Campylobacter is not surprising considering the relatively low number of isolates of each species from each of the geographic locations. While the authors want to make a scientific point of this there is no statistical power to these studies and this can be minimized or removed from the manuscript.	In the revised manuscript, we have minimized the emphasis on serotype abundance.
8	Figure 1 is so small that it is currently unreadable without significant expansion on the screen. It should be rotated to landscape and expanded. The same can be said for many of the figures.	The figures have been improved in the revised manuscript. All figures have been saved in high resolution (dpi>300) and uploaded as figures in the re-submission.
Reviewer #2		
9	In this study, Thystrup et al. report some really valuable and interesting findings about the prevalence of food-borne pathogens across four African countries, investigating the prevalence across different hosts and environments using both whole genome sequencing and shotgun metagenomics. As studies and data collected in low- and middle-income countries are scarce, the data produced would be useful for future research and integration in surveillance databases. I applaud the choices done by the research team by performing, when possible, the research in the countries in which each part of the study was conducted, and the fact that the data obtained by their analyses was reported back to the origin. I have some questions and remarks on both the methodology used to conduct the study and on the results obtained.	Thank you for the kind comment.

10	The authors stated in the methods section that the main subjects of study were children aged between 0 and 5 that were seeking medical treatment for diarrhea. Have the authors observed any changes in the composition of the microbiome between different age differences or differences between how severe the diarrhea was or the duration of the diarrhea? Have you found any differences in serotype prevalence and microbiome composition between individuals living in an urban or rural setting?	Unfortunately, it was not possible to trace questionnaire data (e.g., age, diarrhea severity, or duration) back to most individual samples, which limited our ability to perform such comparisons. This same limitation also prevented us from linking metagenomic and WGS data at the individual level. While we recognize the value of these analyses, the lack of consistently linked metadata was a constraint in this study. We see this as an important consideration for future surveillance efforts, and one we aim to address in subsequent work.
11	Also, I know that maybe the sample size does not allow you to do this type of analysis, but if both the children and the caretaker or sibling were exposed to the same source of FBP, and only the children is experiencing diarrhea, can you identify the same serotype in its caretaker/sibling?	The suggestion is excellent, but unfortunately, the sample size was too limited to explore this type of analysis meaningfully, but we were unable to directly link specific caretakers or siblings to individual diarrheal cases. However, we agree that such paired analyses would be valuable in understanding the transmission dynamics, and hope that future studies with larger, linked datasets will be able to address this question.
12	Regarding the methodologies used, the collected samples were processed, prepared and sequenced using different kits and technologies, depending on the local availability, and as the authors discuss later in the paper, this could be a source of data inconsistencies. However, in the analyses this does not seem to be addressed since I do not see any batch effect correction. Have you looked at whether the different combinations of methods used yielded different quality of data, or whether the kit used were biased towards certain taxa?	Due to the use of multiple sequencing platforms and library preparation kits, it was challenging to apply a traditional batch correction, also due to the low number of samples per batch and the lack of overlapping methodologies. For the WGS data, the aim was to recover high-quality genomes of specific pathogens and understand their genomic relationship with each other, and therefore, we did not expect major biases in genome recovery due to differences in the library prep kits or sequencing methodologies. However, for the metagenomic data where compositional bias is a concern, we investigated the impact of library preparation using PCR free and non-PCR free kits. As described in the Methods section (line 198-200) we compared samples prepared with and without PCR-free kits using a PERMANOVA test and found no statistically significant difference in taxonomic composition. Since taxonomic profiling was

		the primary focus for this part of the study, we felt it was acceptable to pool these samples while acknowledging this as a limitation.
13	It is stated that WGS isolates were assembled using SPAdes, but there is no mention whether the assembled genomes underwent a successive quality control to assess how good the assemblies were. Did the authors try to perform metagenomic assembly on the 158 metagenomic samples or tried to identify virulence factors from either the raw reads of MAGs?	The WGS isolates were indeed assembled using spades, but also underwent subsequent quality control, which is now described in more details in the revised Methods section (line 208-218). Initially, metagenomic assembly was not pursued because tools like KMA and Kraken2 operate directly on raw reads and do not require assembled genomes, but following the suggestions from multiple reviewers, we performed additional analyses which included assembly and binning of MAGs from the metagenomic samples. Virulence factors were then identified using ABRicate.
14	The authors report the prevalence of the WGS isolates, but there is no supplementary table describing the species identified in the metagenomes.	In the revised manuscript, we have added a figure (Figure 4) showing the top 15 most abundant genera identified in the metagenomic samples. We also include the number of genera identified in the metagenomic samples (line 560-563).
15	What type of databases were used with KMA to perform the characterization of the reads? I assume from the description that genbank or refseq was used, but I do not understand why the plural form of database was used. In addition, were the WGS isolates characterized using the assembled genomes or the raw reads?	The database used was a collection of complete and draft genomes from bacteria, archaea, protozoa, fungi, plasmids and viruses, downloaded from GenBank. This has been specified in the revised manuscript (line 243-246). The WGS isolates were characterized using the raw reads.
16	Figure 1 highlights the prevalence of some pathogenic species across the different types of samples, from the y-axis I see that the abundances sums up to 100%, but the text mentions that e.g. E.coli abundance is 4.14, so I assume that the relative abundances were re-normalized when plotted in figure 2, correct? If so, I would mention this particular in the caption since the figure could be a little misleading, or I would just plot the original	Since the figures have been updated, this specific comment may no longer apply directly. However, we fully took the reviewer's suggestion into account when revising this and other related figures.

	values and collapse all the other taxa under an “Other” category.	
17	How the changes in abundances across time and between countries were assessed? There is no statistical test reported in the text, and Figure 2 shows only the average between the countries. Also, what do you think is the reason behind the changes in average abundances in 2021? Could COVID infection be a possible reason?	In the revised manuscript, we updated the temporal analysis to show both total abundance and the abundance of individual pathogens over time. Since this analysis was exploratory and descriptive in nature, we did not apply statistical tests. We agree that this is a relevant hypothesis, and while we cannot confirm causality, we acknowledge this possible influence of pandemic-related disruptions in the discussion (line 925-932).
18	The authors compared the alpha diversity between samples using a subset of species to find differences between countries and sample sources, and for doing that the Kruskal Wallis test was used. The method says that a post-hoc test was used as well, but the pairwise comparisons were not reported in the text.	In the revised manuscript, we replaced the Kruskal-Wallis test with a more appropriate statistical method for our hypothesis and data structure. Additionally, because we applied CLR (centered log-ratio) transformation to the abundance data, we decided to remove the alpha diversity analysis altogether from the updated version.
19	What the ellipses' colors indicate in the ordination plots? Figure 5 has multiple ellipses with the same color. Are they indicating the 95% CI? Has it been tested for differences in composition between dry season and rainy season samples and how intragroup dispersion varies between the seasons?	The plot in question has been replaced with a biplot in the revised manuscript, so the issue regarding the ellipses and their colors is no longer applicable. Regarding seasonality, we originally classified samples using data from the World Bank Climate Change Knowledge Portal. However, after reviewing the sample distribution, we found that most of the high-quality metagenomic samples were collected during the dry season. Because of this imbalance, we decided that a seasonal comparison would not be meaningful and have therefore removed this analysis from the revised manuscript.
20	L101 has a capitalized species name, L453,454,465 Salmonella is not italicized	These errors have been changed in the revised manuscript.
21	The citation for vegan is missing	In the revised version of the manuscript, vegan is no longer used, so this is not applicable anymore.
22	Legend colors are not consistent across the figures	This has been changed in the revised figure.
23	Figure 3 heatmap bottom annotation is missing the labels	This has been changed in the revised figure.

	Reviewer #3	
24	The authors present a collection of 625 WGS isolates and 168 metagenomic samples longitudinally collected from Ethiopia, Mozambique, Nigeria and Tanzania, with the goal of profiling the diversity, composition and genetic context of foodborne pathogens. The study of foodborne pathogens in African countries is of relevance, and this project is well powered and spanned across several countries. The text is clear. However, while the proposed collection is very relevant for the field, most of the presented results are descriptive, with moderate indications of transmission events. In addition, there are widespread issues with the figures. Overall, in the current version, the figures and overall findings presented in this study are not up to the standards of the journal.	We thank the reviewer for this constructive comment. Upon revisiting the manuscript, we fully agreed that the original version was largely descriptive and lacked the analytical depth expected for this journal. In the revised manuscript, we have significantly strengthened the analysis pipeline, including more detailed genomic comparisons, phylogenetic analyses, and additional interpretation of transmission-related patterns. We have also revised and improved all figures for clarity and relevance. In particular, we approached the question of metagenomics for surveillance from a more rigorous angle, aiming to evaluate its feasibility using insights drawn from our unique dataset. We hope these substantial revisions provide a more meaningful and biologically grounded contribution to the field, and better reflect the potential of genomic surveillance in sub-Saharan Africa.
25	The numbers of the figures are all wrong, and they do not coincide with the ones in the main text. The figures often contain formatting issues, are of difficult interpretation and plots are not aligned within the panel. In particular:	Changes have been made to all figures, ensuring that the formatting is correct.
26	Figure 1 (or 4): The plots are not aligned, the bars of different width makes the plots difficult to compare across reservoirs. I recommend to modify this figure into a single horizontal line strip, with all barplots side-by-side, and metadata on top to indicate country of sampling and reservoir type. The authors can also choose to improve the figure in alternative ways, as long as the issues mentioned above are addressed. In addition, as this is the first figure of	A new figure providing an overview of the study design has been added to the manuscript, as suggested by the reviewers. In addition, the reviewers' recommendations for improving the original plot were incorporated into a revised version that presents similar results (Figure 2).

	the manuscript, it should first provide an overview of the study design: displaying the total number of samples collected (for isolation and for metagenomic sequencing), their numbers for each country and each year, the reservoir investigated and the technologies used to do so.	
27	Figure 2 (or 5): As the study sampled at best n=4 African countries, the maps showing the whole continent are predominantly grey. This plot could be a barplot, with each bar representing a country. A barplot would allow to better distinguish the values, and better compare the values from the same country longitudinally. This figure could be included with the previous one.	This information from this figure has now been incorporated into a new figure (Figure 2) so this comment is no longer applicable.
28	Figure 3 (or 6): Species names should always be italicized, clustering of rows and columns can be adjusted to occupy less space, leaving more space for the heatmap and improving readability. Metadata strips should be bigger.	This has been changed in the revised figure.
29	Figure 4 (or 7): Text on top of the subplots has formatting issues and extra spaces, subplots are not aligned. Boxplots of different widths are difficult to compare. Similarly to what suggested for Fig 1, and what done by the authors in Supplementary Figure S5, this figure would be greatly improved if the boxplots would be showed all in one line, next to each other and with equal widths. Legend is missing. It might help to add jitter data to better convey the data point distribution within the boxplots; outlier values should not be displayed (rhombuses).	This figure has been omitted in the new manuscript, since new plots with more meaningful conclusions have been added instead.

30	Figure 5 (or 8): The use of shaded areas covers the position and color of the underlying data points. Shaded areas should be substituted by transparent areas with a colored outline (and only the top 4-5 isolation sources to avoid cluttering).	This has been changed in the revised biplots.
31	Figure 6 (or 9): similar comments to the previous figure: shaded areas on the top PCA make it hard to see the data points. Also, size of the data points should be increased.	This has been changed in the revised biplots.
32	Figure 9 and table 5 are about seasonality, yet seasonality is not mentioned in the Discussions, nor the in the Methods. How are the dry vs wet seasons defined? Is there variability between the sampled countries? In addition, date of collection and associated seasonality are not reported as metadata in the Project ID on ENA.	The plot in question has been replaced with a biplot in the revised manuscript, so the issue regarding the ellipses and their colors is no longer applicable. Regarding seasonality, we originally classified samples using data from the World Bank Climate Change Knowledge Portal. However, after reviewing the sample distribution, we found that most of the high-quality metagenomic samples were collected during the dry season. Because of this imbalance, we decided that a seasonal comparison would not be meaningful and have therefore removed this analysis from the revised manuscript.
33	The analyses presented in the study are all quite high-level, with no in-depth dive into the genomic content of the isolated genomes. One way the authors could provide more functional context to this work is by comparing their foodborne pathogen isolated genomes with public sequences, identify mutations, structural rearrangements, carriage of antimicrobial resistance genes, reconstruct their functional potential. Depending on the sequencing depth (if above >5M reads after preprocessing), strain-level analysis could be performed on the metagenomic samples, which in this case would be extremely relevant as strains belonging to the same species can range from commensal to pathogenic.	We really appreciate the reviewer's insightful suggestions. We agree that the original manuscript lacked deeper genomic analysis, so in the revised version we expanded our analyses in several ways. For the isolate genomes, we conducted phylogenetic analyses, sequence type (ST) assignment, serotyping, and detection of antimicrobial resistance (AMR) genes and virulence factors using standardized tools. We also compared isolate genomes with recovered metagenome-assembled genomes (MAGs) using average nucleotide identity (ANI) and SNP-based phylogenies to assess overlap and relatedness. For the metagenomic data, we performed assembly and binning to recover MAGs, followed by quality assessment, taxonomic classification, and detection of AMR and virulence genes.

	Strain level-information could inform if there are multiple strains of the same species present in the metagenomic samples, providing information on their competition within the host.	While we agree that strain-level profiling from metagenomes could be valuable, most of our metagenomic samples had few pathogens at high enough abundance to allow for strain-level resolution. Additionally, as mentioned in the revised manuscript, there was little to no direct overlap between the isolates and the metagenomic samples due to how the data was originally collected. This limited the feasibility of doing isolate-metagenome strain tracking, though it remains an important direction for future studies.
34	Line 98: I would recommend citing a landmark paper in the use of metagenomics to track out outbreaks: Loman NJ, Constantinidou C, Christner M, Rohde H, Chan JZ, Quick J, Weir JC, Quince C, Smith GP, Betley JR, Aepfelbacher M, Pallen MJ. A culture-independent sequence-based metagenomics approach to the investigation of an outbreak of Shiga-toxigenic Escherichia coli O104:H4. JAMA. 2013 Apr 10;309(14):1502-10. doi: 10.1001/jama.2013.3231. PMID: 23571589.	The paper has now been cited in the Introduction and incorporated into the Discussion section as well. We were inspired by the methodology presented in this study and have drawn from it in our revised analyses, so thank you for the helpful suggestion.
35	Legend of figure 1 (or 4): it should specify “foodborne pathogen”	This has been changed in the revised figure.
36	Legend of Figure 2 (or 6): the metadata strips should also include information on the reservoir type (human, animal, waste), similarly to what done for Figure 1; “CLR-value” should be extended to changed to CLR transformed abundance.	This has been changed in the revised figure.
37	Legend of Figure 5 (or 8): the metadata strips should also include information on the reservoir type (human, animal, waste), similarly to what done for Figure 1;	This has been changed in the revised figure.
38	Legend of figure 5 (or 8): it should specify “Isolation source”, instead of “iso_source”.	This has been changed in the revised figure.

39	Supplementary Table S6 is too big to be provided as .doc document; it should be provided as .csv or .tsv, or excel.	This has been changed and will be supplied as a .xlsx file in the revised manuscript.
40	Targeted sequencing depth should be included in the methods sections.	We have now included the targeted sequencing depth in the Methods section, based on the guidelines provided by Gweon et al. (2019).
41	For metagenomic samples, the number of microbial reads after pre-processing should be included in the methods section.	This has been added to the Methods section in the revised manuscript.

	Reviewer comment	Response
	Reviewer #2	
1	The authors have addressed all the raised points, I have no further comments.	Thank you for addressing the edits and confirming that the manuscript has been improved with the changes.
	Reviewer #3	
2	The authors have addressed all of my previous comments, including the addition of the suggested analyses and the inclusion of more biologically relevant results. My only remaining suggestion is to rotate Figure 3 so that the taxa names appear as rows instead of columns, which would improve readability.	Thank you for addressing the edits and confirming that the manuscript has been improved with the changes. Figure 3 will be rotated so that the taxa names will appear as rows. Also, the editor has suggested to clarify in Figure 1 that the samples analyzed were from different sources, so we have tried to make that clear in the revised manuscript.
	Reviewer #4 (mediator for reviewer #1)	
3	As the mediator stepping in for Reviewer #1, my remit is to assess whether the authors have satisfied that reviewer's main concerns. The ancillary matter of data availability is largely resolved, although the genome assemblies themselves have not yet been released.	Thank you for pointing out that the issues with the assemblies have not been solved. We have made sure that the assemblies are now available on ENA together with the rest of the data, as well as provided the individual accession numbers for the assemblies in the Supplementary Table-sheet.
4	The second—and far more substantial—issue, which I share, concerns the claimed integration of metagenomic and isolate WGS data. The manuscript is still framed as an “integrative” analysis in which metagenomic sequencing corroborates or augments isolate-based whole-genome surveillance, yet the authors now acknowledge that the metagenomic and WGS datasets were generated from entirely different specimens. Without paired samples, the revised analyses amount to incidental overlaps: the metagenomic data show no strain-level concordance, uncover no biologically meaningful	To address the concerns of the point regarding the use of the word “integrative analysis”, the editor has suggested that we revise the title and change the framing of the paper to remove references to it being an “integrated” analysis. This included rewording some sections of the Methods and Results to make it clear that

	findings (e.g., novel lineages or strain carriage patterns), and yield no epidemiologically useful insight. In short, the metagenomic component can no longer sustain the study's unique selling point, leaving the central claim unsupported. Rectifying this gap would require either generating a new, paired specimen set—clearly beyond the scope of a revision—or removing the metagenomic half of the study and rewriting the narrative accordingly, which would markedly dilute the manuscript's claimed novelty and impact. Given these constraints, Reviewer #1's main concern remains unresolved.	the analyses were not performed on paired samples. Please refer to lines 164-168 and 279-281 for details). It has also been emphasized in the Discussion that this is a limitation (please see lines 423-425 and 577-583 for details).
--	--	--